# TEXT-TO-3D BY STITCHING A MULTI-VIEW RECONSTRUCTION NETWORK TO A VIDEO GENERATOR

**Hyojun Go**[1]    **Dominik Narnhofer**[1]    **Goutam Bhat**[2]    **Prune Truong**[2]
**Federico Tombari**[2]    **Konrad Schindler**[1]
[1]ETH Zurich, [2]Google

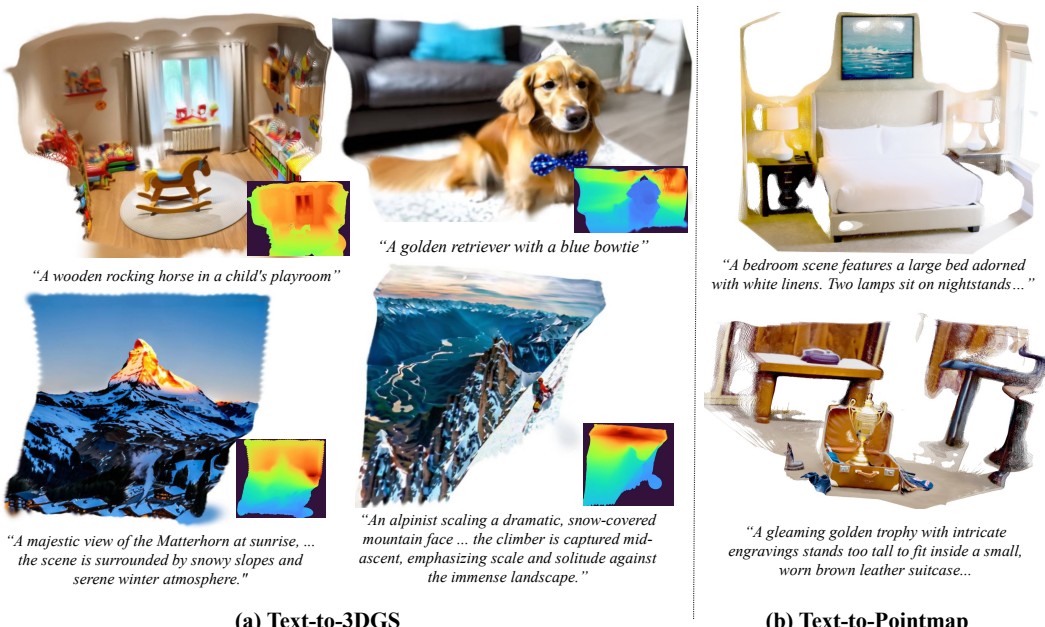

**(a) Text-to-3DGS**                **(b) Text-to-Pointmap**

Figure 1: **Text-to-3D generation with *VIST3A*.** Video models excel at generating latent visual content from text prompts, whereas 3D foundation models shine when it comes to decoding such a latent representation into consistent scene geometry. By stitching a video generator and a 3D reconstruction network together and aligning their latents, we obtain an end-to-end model that produces high-quality Gaussian splats (a) or point maps (b) from text prompts.

## ABSTRACT

The rapid progress of large, pretrained models for both visual content generation and 3D reconstruction opens up new possibilities for text-to-3D generation. Intuitively, one could obtain a formidable 3D scene generator if one were able to combine the power of a modern latent text-to-video model as "generator" with the geometric abilities of a recent (feedforward) 3D reconstruction system as "decoder". We introduce VIST3A, a general framework that does just that, addressing two main challenges. First, the two components must be joined in a way that preserves the rich knowledge encoded in their weights. We revisit *model stitching*, i.e., we identify the layer in the 3D decoder that best matches the latent representation produced by the text-to-video generator and stitch the two parts together. That operation requires only a small dataset and no labels. Second, the text-to-video generator must be aligned with the stitched 3D decoder, to ensure that the generated latents are decodable into consistent, perceptually convincing 3D scene geometry. To that end, we adapt *direct reward finetuning*, a popular technique for human preference alignment. We evaluate the proposed VIST3A approach with different video generators and 3D reconstruction models. All tested pairings markedly improve over prior text-to-3D models that output Gaussian splats. Moreover, by choosing a suitable 3D base model, VIST3A also enables high-quality text-to-pointmap generation.

Project page: https://gohyojun15.github.io/VIST3A/

# 1 INTRODUCTION

With image and video generators now a commodity, text-to-3D models that produce 3D scenes from text prompts have become a new research frontier, with applications in AR/VR, gaming, robotics, and simulation. Early methods for 3D generation adopt Score Distillation Sampling (SDS) (Poole et al., 2023; Tang et al., 2024b; Wang et al., 2023b; Chen et al., 2024b) to optimize a 3D representation, e.g. a NeRF (Mildenhall et al., 2021; Müller et al., 2022) or 3D Gaussian Splats (3DGS, Kerbl et al., 2023) under a pretrained 2D diffusion prior (Rombach et al., 2022). A drawback these methods have in common is the need for slow per-scene optimization. Another line of work uses multi-stage pipelines that first synthesize images and then lift them to 3D with a separate model (Tang et al., 2024a; Xu et al., 2024b; Zhang et al., 2024b) or with per-scene optimization (Gao et al., 2024; Wu et al., 2024a; Yu et al., 2025b); employ progressive warping and refinement (Shriram et al., 2025; Yu et al., 2025a; 2024); or sequentially chain multiple generative modules (Yang et al., 2025b; Engstler et al., 2025). The multi-stage design not only increases model complexity and engineering effort, but also makes such models prone to error accumulation (Lin et al., 2025; Meng et al., 2025).

A recent trend is to directly generate the 3D representation with end-to-end latent diffusion models (LDMs, Schwarz et al., 2025; LAN et al., 2025; Li et al., 2025b;a). A prominent line of work starts from pretrained 2D image (Esser et al., 2024; Rombach et al., 2022) or video models (Team, 2024; Yang et al., 2025e) and finetunes them to output multi-view 2D latents, reusing the pretrained priors (Szymanowicz et al., 2025; Liang et al., 2025; Schwarz et al., 2025; Lin et al., 2025; Yang et al., 2025c; Go et al., 2025a;b). Subsequently, a VAE-style decoder is trained to decode those latents into the desired 3D representation, see Fig. 2. The LDM-like design unifies 2D generation and multi-view reconstruction within the latent space and enables efficient 3D scene generation with a compact, well-amortized decoder.

Figure 2: **Comparison with existing, LDM-based 3D generators.** Instead of training a custom decoder from multi-view 2D latents to 3D outputs, we stitch and align an existing, pretrained 3D reconstruction model.

Still, two key limitations remain. First, we argue that the Achilles heel of existing 2D-to-3D diffusion models is the decoder. By simply repurposing the 2D VAE to produce 3D outputs, the network must learn 3D reconstruction more or less from scratch, which requires extensive training and large datasets that are hard to obtain (Yang et al., 2025c; Szymanowicz et al., 2025; Go et al., 2025b). This practice becomes increasingly problematic as new, better 3D foundation models emerge (Wang et al., 2026; 2025b; 2024; Zhang et al., 2025) and the ad-hoc trained decoders of text-to-3D models fall further behind the state of the art in 3D vision.

Second, the prevalent training scheme tends to suffer from weak alignment between the generative model and the VAE decoder. Typically, the former is finetuned on multi-view datasets with a generative objective like a diffusion loss (Song et al., 2021; Sohl-Dickstein et al., 2015; Ho et al., 2020) or flow matching (Liu et al., 2023; Lipman et al., 2023; Albergo & Vanden-Eijnden, 2023), which only indirectly promotes 3D-consistent latents. Moreover, the separate training may cause the latents, even if 3D-consistent, to be out of domain from the perspective of the decoder. To mitigate that misalignment, it has been proposed to add rendering losses that promote decodable latents (Lin et al., 2025). However, the resulting objective is based on single-step sampling and does not sufficiently take into account the denoising trajectory, leading to weak alignment at inference.

We introduce **VIST3A**: **VI**deo VAE **ST**itching and **3D A**lignment. The proposed method consists of two complementary components that address the above-mentioned limitations, see Fig. 2. First, we resort to the concept of *model stitching* (Pan et al., 2023; Lenc & Vedaldi, 2015; Bansal et al., 2021; Csiszárik et al., 2021; Yang et al., 2022) to leverage powerful, pretrained feedforward 3D models for decoding, rather than start from scratch. The idea is to attach the relevant part of a 3D reconstruction network as a "decoder" to the latent space of a video VAE. For this to work, there needs to be one or more layers in the 3D model whose activations are similar (up to a linear transformation) to those in the VAE's latent space, despite their independent pretraining. Perhaps surprisingly, this turns out to be the case. For the 3D model, we identify the layer with the most linear relation to the LDM latents,

slice the network before that layer, and retain the downstream portion as 3D decoder. After fitting a single, linear stitching layer (in closed form), the VAE latent space already matches the expected input of the 3D decoder well, such that subsequent fine-tuning will be minor and not degrade the respective generative and 3D reasoning capabilities of the two base models.

Second, we further improve alignment between the generative model and the stitched decoder through *direct reward finetuning* (Clark et al., 2024; Xu et al., 2023; Prabhudesai et al., 2024; Wu et al., 2024c; Shen et al., 2025). In that technique, commonly used to align diffusion models with human preferences, reward signals are defined based on the "goodness" of the VAE output – in our setting, the visual quality and 3D consistency of the decoded 3D representations. Maximizing these rewards encourages the LDM to produce latents that are 3D-consistent and lie within the decoder's input domain, ensuring high-quality outputs. Importantly, our alignment compares video model outputs and images rendered from the generated 3D scenes, hence it does not require labels.

In our experiments, we show that the proposed stitching scheme is applicable across a range of video generative models and also across several different feedforward 3D models. VIST3A's direct 3D decoding consistently outperforms prior text-to-3DGS methods, and additionally offers high-quality pointmap generation from text prompts.

## 2 RELATED WORKS

**3D generation.** Recent works have explored various 3D representations for generative modelling, including point clouds (Mo et al., 2023; Nichol et al., 2022; Vahdat et al., 2022), meshes (Xu et al., 2024a), voxel grids (Sanghi et al., 2023), NeRFs (Chen et al., 2023; Müller et al., 2022; Mildenhall et al., 2021), and 3DGS (Henderson et al., 2024; Zhang et al., 2024a; Kerbl et al., 2023). Score distillation using 2D diffusion models is time-consuming, as it requires per-scene test time optimization (Wang et al., 2023a; Shi et al., 2024; Wang et al., 2023b), while multi-stage pipelines (Yu et al., 2025b; Liu et al., 2024; Zheng et al., 2025) lack robustness and create significant engineering overhead. For further details on multi-stage pipelines, please refer to Appendix A.

More recently, the field has shifted towards end-to-end latent diffusion models, where the generator operates in the latent space of a VAE, and the latter directly decodes the resulting latents to 3D outputs. Many of these works focus on object-centric asset generation (Wu et al., 2024b; Zhao et al., 2023; Lin et al., 2025) and train the LDM on curated datasets such as Objaverse (Deitke et al., 2023), with single objects or bounded scenes, and controlled camera paths. Consequently, they are unable to handle real-world challenges like strongly varying scene scale, variable lighting, etc.

To tackle such situations, recent methods (Szymanowicz et al., 2025; Liang et al., 2025; Schwarz et al., 2025; Lin et al., 2025; Yang et al., 2025c; Go et al., 2025a;b) repurpose the comprehensive knowledge of the visual world that is implicit in 2D image generators. The general strategy is to finetune a pretrained 2D model on multi-view data, by using generative losses to enforce cross-view consistency. In many cases training is further supported by additional 3D cues like camera poses (Li et al., 2024b; Go et al., 2025b), depthmaps (Go et al., 2025a; Yang et al., 2025c), or pointmaps (Szymanowicz et al., 2025). The resulting multi-view latents are decoded to 3D scenes with a dedicated VAE-style decoder, meaning that 3D reasoning capabilities must be rebuilt from scratch, and that they are only weakly aligned with the generator output – limitations which we address with VIST3A.

**Learned 3D reconstruction.** A notable trend in 3D computer vision is the trend to move away from multi-stage pipelines and iterative optimization towards end-to-end, feedforward 3D modelling. Classical reconstruction pipelines based on SfM (Hartley & Zisserman, 2003; Schönberger & Frahm, 2016) and MVS (Furukawa et al., 2015; Schönberger et al., 2016) require incremental, iterative optimization, whereas recent advances like DUSt3R (Wang et al., 2024) and MASt3R (Leroy et al., 2024) directly predict 3D point maps in one forward pass. Several follow-up works have further reduced test-time optimization (Tang et al., 2025; Wang et al., 2025c; Yang et al., 2025a). Likewise, 3D Gaussian splatting has evolved from per-scene optimization to feedforward prediction (Charatan et al., 2024; Chen et al., 2024a; Ye et al., 2025). Once more, data scaling has been a critical factor (Wang et al., 2025b; 2026). Consequently, replicating the 3D capabilities of recent feedforward models as part of VAE training would be difficult and costly. VIST3A offers a solution by reusing, rather than rebuilding, models like AnySplat (Jiang et al., 2025), VGGT (Wang et al., 2025b), or MVDUSt3R (Tang et al., 2025).

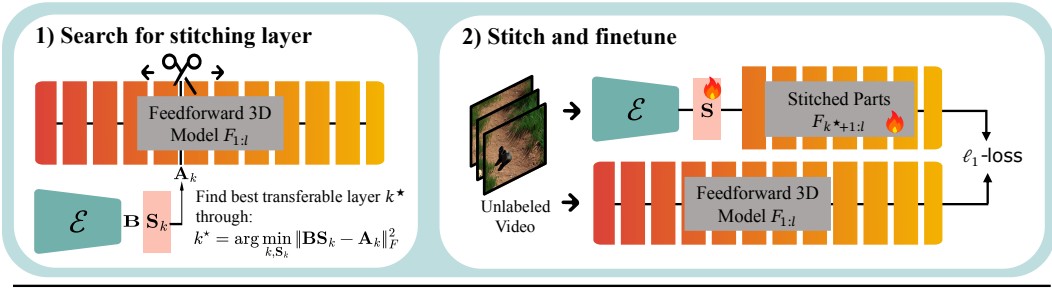

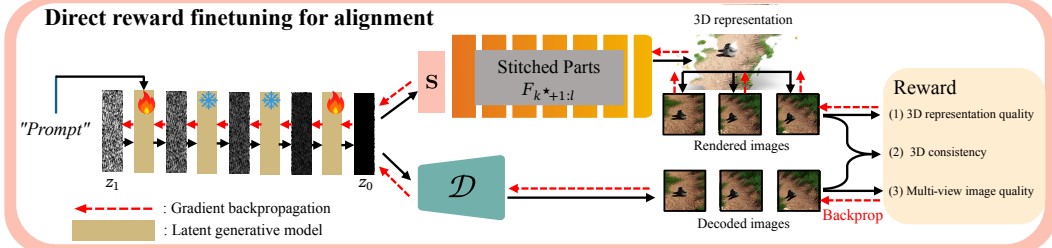

Figure 3: **VIST3A constructs a 3D VAE through model stitching (top), then aligns it with a generative model via direct reward finetuning (bottom).** Stitching repurposes a part of a pretrained 3D vision model as decoder to obtain a 3D VAE. Direct reward finetuning simulates full-trajectory denoising, forcing the generative model to produce 3D-consistent, decodable latents.

**Model stitching.** Recomposing the heads and tails of two different networks was initially studied as a way to assess the equivariance of neural representations (Lenc & Vedaldi, 2015), and as an experimental tool to compare two different representations (Csiszárik et al., 2021; Bansal et al., 2021). To ensure invariance against trivial affine transformations, the head of some trained network $A$ is normally attached to the tail of another network $B$ via a linear, trainable *stitching layer*. Besides revealing similarities between networks that common metrics like CKA (Kornblith et al., 2019) would miss, it was also found that different architectures that were trained on the same data can often be stitched into a new, hybrid model with minimal degradation (Bansal et al., 2021). This has opened the door for practical uses of stitching, e.g. DeRy (Yang et al., 2022) for resource-constrained reassembly of pretrained models and SN-Net (Pan et al., 2023) to build networks with varying scales. Going one step further, we demonstrate that strong 3D VAEs[1] can be obtained by stitching a foundational 3D model to the latent space of a video VAE as its decoder, even if they were trained independently on different data.

## 3 METHODOLOGY

VIST3A consists of two key components, see Fig. 3: (1) model stitching to optimally attach (part of) a foundational 3D model as the decoder for the latent, and (2) direct reward finetuning to optimize the alignment of the (latent) generative model with that new decoder.

### 3.1 MODEL STITCHING FOR 3D VAE CONSTRUCTION

Our objective is to build a 3D VAE by seamlessly combining the encoder of a video LDM and a feedforward 3D reconstruction model. Note that, for stitching purposes, one can skip the denoising loop, since feeding images into the encoder already gives clean latents. Let $\mathcal{E}$ denote the encoder and $\mathcal{D}$ the decoder of the VAE, and let $F_{1:l}(\boldsymbol{x}) = f_l \circ \cdots \circ f_1(\boldsymbol{x}) = \boldsymbol{y}$ be the feedforward 3D network that maps a set of views $\boldsymbol{x}$ to a 3D output $\boldsymbol{y}$, with $l$ the total number of layers in that feedforward model. As shown in Fig. 3, we cut the feedforward model at layer $k^*$ and stitch the downstream part $F_{k^*+1:l} = f_l \circ \cdots \circ f_{k^*+1}$ to the output layer of the encoder $\mathcal{E}$, with the help of a linear *stitching layer* $\mathbf{S}$. In doing so, we obtain a new 3D VAE $\mathcal{M}_{\text{stitched}}$ that outputs the same representation $\hat{\boldsymbol{y}}$ as the original 3D model:

$$\mathcal{M}_{\text{stitched}} = F_{k^\star+1:l} \circ \mathbf{S} \circ \mathcal{E}(\boldsymbol{x}) = \hat{\boldsymbol{y}}, \quad \mathcal{D}_{\text{stitched}} = F_{k^\star+1:l} \circ \mathbf{S} \tag{1}$$

---

[1]To be consistent with existing literature (LAN et al., 2025; Yang et al., 2025c), we also use the term "3D VAE", although the mapping from 2D images to 3D scene is, technically, not a variational auto-encoder.

The front portion $F_{1:k^\star}$ of the 3D model is discarded – but if the clean encoder latents, after the affine warping $\mathbf{S}$, are (almost) the same as the activations $f_{k^\star}$, then the back portion will still produce the same output, $\hat{\boldsymbol{y}} \approx \boldsymbol{y}$. In other words, the stitched VAE $\mathcal{M}_{\text{stitched}}$ is an approximation of the original 3D model $F$. It retains much of the ability to map multi-view images to a 3D reconstruction and only requires a little fine-tuning to restore that ability.

**Step 1: Finding the stitching index and initialization.** To identify the layer $k^*$ in the 3D model whose representation is most compatible with the VAE latent, we first push a set of $N$ samples through the encoder $\mathcal{E}$ to obtain their latents $\mathbf{B} \in \mathbb{R}^{N \times D_\mathcal{E}}$. Here, $D_\mathcal{E}$ denotes the dimensionality of the encoder latent space, and $D_F^k$ denotes the dimensionality of the feature (activation) at layer $k$. Then, we scan over candidate layers $k \in \{1, ..., l-1\}$ of the 3D model and, for each layer in turn, extract the activations $\mathbf{A}_k \in \mathbb{R}^{N \times D_F^k}$ and fit the linear stitching layer $\mathbf{S}^*_k \in \mathbb{R}^{D_\mathcal{E} \times D_F^k}$ that best recovers the activations of the 3D model at layer $k$, by solving a least-squares problem:

$$\mathbf{S}^*_k = \arg\min_{\mathbf{S}_k} \|\mathbf{B}\mathbf{S}_k - \mathbf{A}_k\|_F^2 = \left(\mathbf{B}^\top \mathbf{B}\right)^{-1} \mathbf{B}^\top \mathbf{A}_k. \tag{2}$$

Finally, we select the stitching layer $k^\star$ that leads to the smallest (mean squared) error, $k^\star = \arg\min_k \|\mathbf{B}\mathbf{S}_k^* - \mathbf{A}_k\|_F^2$, and assemble the 3D VAE by concatenating $\mathcal{E}$, $\mathbf{S}^*_{k^\star}$ and $F_{k^\star+1:l}$. Empirically, we find that many combinations of foundational VAEs and 3D feedforward models can be stitched in this manner, with minimal performance loss.

**Step 2: Stitched decoder finetuning.** To further reduce the remaining discrepancies between the newly assembled 3D VAE and the original 3D model, we finetune $\mathbf{S}$ and $F_{k^\star+1:l}$ to reproduce the predictions of the original 3D model $\boldsymbol{y}$, using them as pseudo-targets. Practical feedforward models produce multiple outputs (e.g., point maps, depth, poses), so we optimize a weighted sum of $\ell_1$ losses for all of them. Note that the fine-tuning step is self-supervised and does not require labels. In our implementation, we restrict the stitching layer to a 3D convolution and employ LoRA (Hu et al., 2022) for updating $F_{k^\star+1:l}$, to prevent large deviations from the pretrained weights. For further details, see Appendix B.1.

## 3.2 Alignment via Direct Reward Finetuning

So far, we have assembled a 3D VAE with a strong, pretrained 3D decoder. However, during text-to-3D inference, the latents are not obtained from the encoder but generated from noise by the denoising loop conditioned on the text prompt. Therefore, we must also align the generative model itself with the 3D decoder, such that it produces decodable latents.

Previous work finetunes the generative network by minimizing generative losses over some multi-view dataset. Unfortunately, that strategy does not ensure 3D-consistent latents. Even if it did, the finetuning bypasses the decoder, hence there is no guarantee that the generated latents fall within the distribution expected by the 3D VAE and can be decoded to meaningful outputs.

To address the disconnect between the denosing loop and the 3D VAE, we adopt direct reward finetuning to align the two. In other words, we extend conventional, generative multi-view finetuning with reward maximization. The conventional generative loss $L_{\text{gen}}$ uses paired data, i.e., multi-view images and corresponding prompts. In contrast, the proposed reward term $r(\cdot, c)$ relies only on the text prompt and requires no ground-truth images. Our total loss is defined as

$$L_{\text{total}} = L_{\text{gen}} - r\big(z_0(\theta, c, z_T), c\big), \tag{3}$$

where $\theta$ are the parameters of the video generative model, $c$ represents the text prompt, $z_T$ is the initial noise, and $z_0(\theta, c, z_T)$ is the final latent produced by the denoising loop.

**Reward.** The proposed reward function consists of three components that ensure high-quality and 3D-consistent generation. *(1) Multi-view Image Quality:* As we keep the encoder frozen, the generated latents can be decoded by the original video decoder $\mathcal{D}$ to obtain multi-view images. We evaluate these images against the input prompt using CLIP-based (Fang et al., 2024) and HPSv2 human preference scores (Wu et al., 2023) to promote prompt adherence and visual quality, similar to DanceGRPO (Xue et al., 2025). *(2) 3D Representation Quality:* To encourage high-quality 3D outputs after decoding with $\mathcal{D}_{\text{stitched}}$, we render the generated 3D scenes (pointmaps and/or 3DGS) back into 2D views and apply the same (CLIP + HPSv2) metrics to them as above. *(3) 3D Consistency:* To enforce 3D consistency, we render the 3D representation from the same viewpoints as the multi-view images reconstructed by the video decoder $\mathcal{D}$, using the camera poses predicted by the

feedforward 3D model. We then compute a combination of $\ell_1$-loss and LPIPS (Zhang et al., 2018) for each pair of decoded and rendered images belonging to the same viewpoint. The final (negative) reward is a weighted sum of these three losses. For further details, see Appendix B.2.

**Alignment algorithm.** To optimize the generative model according to the reward function above, we employ direct reward finetuning (Clark et al., 2024; Xu et al., 2023; Prabhudesai et al., 2024; Wu et al., 2024c; Shen et al., 2025). I.e., the model generates samples by unfolding the full denoising path, and the rewards computed from these samples are then backpropagated through the denoising chain. While the algorithm benefits from gradient-based feedback, it can also suffer from exploding gradient norms. To stabilize the optimization, we generalize the idea of DRTune (Wu et al., 2024c): gradients are detached from the inputs to the generative model, but retained during the update step to the next denoising state. In this way, reward propagation remains stable even at early denoising steps. Furthermore, we modify the optimizer for better computational efficiency by (i) randomized sampling, using fewer timesteps than during inference, and (ii) randomizing the subset of denoising steps where gradients are backpropagated, such that the model learns from diverse denoising trajectories. For further details, see Appendix B.2.

In summary, we perform joint, end-to-end alignment of the VAE and the generative model, unlike conventional multi-view fine-tuning that keeps them separate. Reward tuning ensures that, throughout the iterative denoising process, the generative model remains aligned with our 3D VAE and generates latents that suit the stitched decoder.

## 4 EXPERIMENTAL RESULTS

In what follows, we demonstrate **VIST3A**'s text-to-3D generation performance. The main findings are that **VIST3A** clearly outperforms existing feedforward text-to-3DGS approaches and also offers high-quality text-to-pointmap generation. Moreover, we experimentally analyze our two core components, self-supervised *model stitching* and *alignment finetuning*.

### 4.1 EXPERIMENTAL SETUPS

We provide a high-level overview of the experimental setup. A complete description of evaluation protocols and training details can be found in Appendix C.

**Target 3D models.** We target last-generation foundational 3D vision models that have been trained on large-scale datasets, have demonstrated generality and reliable performance across diverse domains, and require only images as input. For our experiments, we select three representative state-of-the-art models: *(1) MVDUSt3R* (Tang et al., 2025) predicts pointmaps and Gaussian splats, *(2) VGGT* (Wang et al., 2025b) predicts pointmaps, depth maps and camera poses, and *(3) AnySplat* (Jiang et al., 2025) predicts Gaussian splats and camera poses.

**Target video generators.** Our primary video model is Wan 2.1 T2V large (Wan et al., 2025), a state-of-the-art text-to-video generator. To demonstrate the generality of VIST3A across different architectures, we additionally use several other latent video models, including CogVideoX (Yang et al., 2025e), SVD (Blattmann et al., 2023), and HunyuanVideo (Kong et al., 2024).

**Training data.** We finetune stitched VAEs on DL3DV-10K (Ling et al., 2024) and ScanNet (Dai et al., 2017), without 3D labels. To align the video generator in latent space, we utilize DL3DV-10K to compute the generative loss, with prompts from the HPSv2 training set (Wu et al., 2023).

### 4.2 MAIN RESULTS: 3D GENERATION

Stitching Wan to the 3D models listed in Section 4.1 yields two types of generative models: (i) Text-to-3DGS when using AnySplat or MVDUSt3R as decoder; and (ii) Text-to-Pointmap when using VGGT or MVDUSt3R. Both variants are evaluated in the following.

**Baselines.** Important baselines for text-to-3DGS are SplatFlow (Go et al., 2025a), Director3D (Li et al., 2024b), Prometheus3D (Yang et al., 2025c), and VideoRFSplat (Go et al., 2025b). Additionally, we include Matrix3D-omni (Yang et al., 2025d), to our knowledge, the only other model that unifies generation and reconstruction in latent space.

Table 1: **Quantitative results on T3Bench and SceneBench.**

| Method | T3Bench (Object-centric) | | | | | | SceneBench (Scene-level) | | | | | |
| | Imaging↑ | Aesthetic↑ | CLIP↑ | Unified Reward | | | Imaging↑ | Aesthetic↑ | CLIP↑ | Unified Reward | | |
| | | | | Align.↑ | Coher.↑ | Style↑ | | | | Align.↑ | Coher.↑ | Style↑ |
|---|---|---|---|---|---|---|---|---|---|---|---|---|
| Matrix3D-omni | 43.05 | 37.66 | 25.06 | 2.44 | 3.10 | 2.69 | 46.65 | 37.62 | 24.04 | 2.66 | 3.29 | 2.80 |
| Director3D | 54.32 | 53.33 | 30.94 | 3.25 | 3.43 | 3.05 | 47.79 | 52.81 | 29.31 | 3.36 | 3.67 | 3.20 |
| Prometheus3D | 47.46 | 44.32 | 29.15 | 2.84 | 3.12 | 2.66 | 44.73 | 45.85 | 28.57 | 3.20 | 3.36 | 2.98 |
| SplatFlow | 46.09 | 53.24 | 29.48 | 3.29 | 3.25 | 2.93 | 48.85 | 53.71 | 29.43 | 3.47 | 3.65 | 3.26 |
| VideoRFSplat | 46.52 | 39.50 | 30.13 | 3.12 | 3.24 | 3.09 | 58.19 | 51.71 | 29.76 | 3.58 | 3.63 | 3.30 |
| **VIST3A**: Wan + MVDUSt3R | **58.83** | **56.55** | **32.75** | **3.56** | **3.89** | **3.56** | 62.08 | 55.67 | 30.26 | 3.72 | 3.97 | 3.47 |
| **VIST3A**: Wan + AnySplat | 57.03 | 54.11 | 31.38 | 3.36 | 3.68 | 3.17 | 64.87 | 56.96 | 30.18 | 3.67 | 3.86 | 3.40 |

Table 2: **Quantitative results on DPG-Bench.**

| Method | DPG-Bench | | | | |
| | Global↑ | Entity↑ | Attribute↑ | Relation↑ | Other↑ |
|---|---|---|---|---|---|
| Matrix3D-omni | 53.32 | 42.44 | 56.23 | 37.12 | 10.32 |
| Director3D | 66.67 | 64.96 | 60.85 | 45.15 | 22.73 |
| Prometheus3D | 45.45 | 48.35 | 55.03 | 33.50 | 9.10 |
| SplatFlow | 69.70 | 68.43 | 65.55 | 50.49 | 40.91 |
| VideoRFSplat | 36.36 | 56.93 | 66.89 | 48.53 | 31.82 |
| **VIST3A**: Wan + MVDUSt3R | **81.82** | 84.31 | **86.13** | 68.93 | **54.55** |
| **VIST3A**: Wan + AnySplat | 78.79 | **85.58** | 84.12 | **76.70** | 45.45 |

Table 3: **Stitching enhances NVS.**

| Method | PSNR↑ | SSIM↑ | LPIPS↓ |
|---|---|---|---|
| SplatFlow | 19.10 | 0.671 | 0.278 |
| VideoRFSplat | 19.05 | 0.674 | 0.281 |
| Prometheus3D | 19.56 | 0.683 | 0.277 |
| AnySplat | 20.85 | 0.695 | 0.238 |
| Hunyuan + AnySplat | 21.17 | 0.710 | 0.242 |
| SVD + AnySplat | **21.48** | **0.720** | **0.218** |
| CogVid + AnySplat | 21.32 | 0.716 | 0.222 |
| Wan + AnySplat | 21.29 | 0.718 | 0.232 |

**Evaluation protocol.** We evaluate text-to-3DGS models on three benchmarks: T3bench (He et al., 2023) for object-centric generation, SceneBench (Yang et al., 2025c) for scene-level synthesis, and DPG-bench (Hu et al., 2024) to assess adherence to long, detailed prompts. On T3bench and SceneBench, we render images and compute Imaging Quality and Aesthetic Quality scores as defined by VBench (Huang et al., 2024) to assess visual fidelity, CLIP score (Hessel et al., 2021) for text-prompt alignment, and Alignment, Coherence, and Style scores according to Wang et al. (2025d) as comprehensive quality metrics. We prefer to avoid traditional no-reference metrics like NIQE (Mittal et al., 2012b) and BRISQUE (Mittal et al., 2012a) that have sometimes been used in the context of 3D generation, but lack a meaningful connection to the conditional generation task (e.g., they can be gambled by always returning the same sharp and colorful, high-scoring image, independent of the prompt). For DPG-bench, we follow the suggested protocol (Hu et al., 2024), but upgrade from the originally proposed language models to the more capable, UnifiedReward LLM (based on Qwen 7B). Text-to-pointmap models are evaluated qualitatively, as no established benchmarks or baselines exist.

**Quantitative Results.** Tables 1 and 2 show the results for the three text-to-3DGS benchmarks. Notably, both tested VIST3A variants exhibit superior performance across all datasets and evaluation metrics. On T3bench, both Wan+AnySplat and Wan+MVDUSt3R consistently outperform all baselines, with particularly large margins in Imaging Quality and Coherence score. For the more complex scene-level synthesis of SceneBench, our models reach Imaging Quality scores >60 and Coherence scores >3.8, again a marked improvement over prior art. On DPG-bench, our models greatly outperform the baselines, mostly scoring >75 (often even ≈85), values that previously seemed out of reach. The consistent gains on T3bench, SceneBench, and DPG-bench demonstrate the effectiveness and versatility of our stitching approach for text-based 3D scene generation. We attribute these results to the power of foundational contemporary video and 3D models, which our stitching and fine-tuning scheme unlocks for the purpose of 3D generative modeling.

**Human evaluation.** To further validate the effectiveness of VIST3A, we conduct a user study where we compare it against four other methods: Director3D, SplatFlow, Prometheus3D, and VideoRFSplat. A total of 28 participants evaluated 14 randomly selected samples drawn from T3Bench, SceneBench, and DPG-Bench, ranking each method according to two criteria: (1) Text Alignment and (2) Visual Quality of videos rendered from the generated 3DGS. As shown in Table 4, VIST3A achieves the best performance (lowest average rank) on both criteria. Notably, participants rank VIST3A as the top method in >68% of cases for text alignment and >87% for visual quality, underscoring its superiority in generating high-fidelity, semantically consistent 3D scenes.

**Qualitative Results.** Figure 4 qualitatively compares VIST3A (Wan+AnySplat) to several baselines. In line with the quantitative results, VIST3A produces superior, visually compelling, and geometrically coherent renderings that closely follow the input prompts; whereas previous methods tend to exhibit artifacts, structural distortions, and poor text alignment. Further qualitative results, including Wan+MVDUst3R and Wan+AnySplat variants of VIST3A, as well as text-to-pointmap

Table 4: **User study results.** Participants rank five methods in terms of text alignment and visual quality of rendered videos from generated 3DGS (lower average rank is better).

| Method | Text Alignment (Avg. Rank ↓) | Visual Quality (Avg. Rank ↓) |
|---|---|---|
| Director3D | 3.03 | 2.99 |
| SplatFlow | 3.38 | 3.88 |
| Prometheus3D | 3.25 | 3.71 |
| VideoRFSplat | 2.74 | 2.92 |
| **VIST3A (Wan+ AnySplat)** | **1.54** | **1.45** |

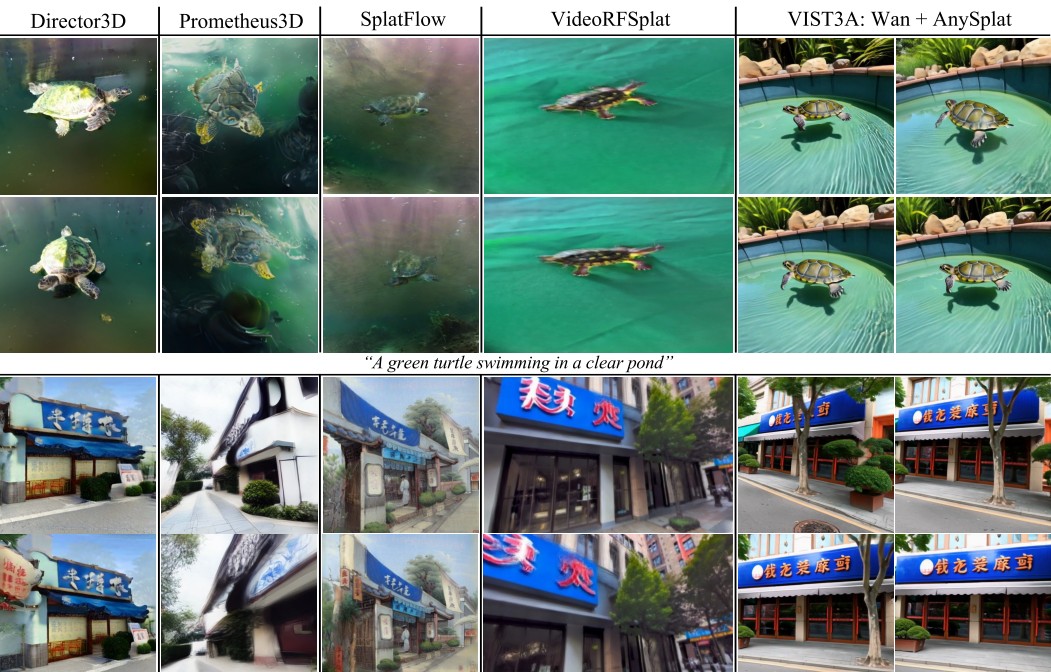

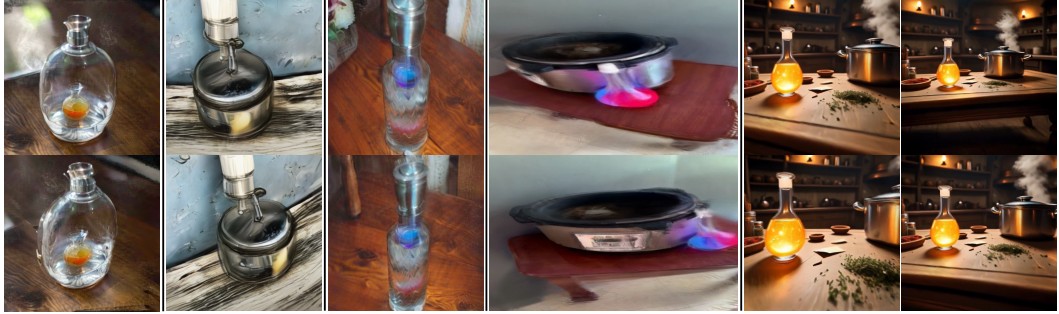

Figure 4: **Qualitative results for 3DGS generation.** We show samples from T3Bench (top), SceneBench (middle), and DPG-bench (bottom). VIST3A generates realistic and crisp 3D scenes and adheres to intricate details in the prompt.

examples, can be found in Appendix E. Interestingly, we find that, even without specific training on very long image sequences, VIST3A can generate coherent large-scale scenes by extending the number of frames generated by the LDM. This demonstrates that our framework preserves the ability of video generator and the 3D decoder to handle long sequences. Examples are depicted in Fig. 16.

Table 5: **Results of point map reconstruction with stitched models.**

| Method | Pointmap Estimation | | | | | | | | | | | | | | | | | Camera Pose Estimation | | | | | |
| | 7-Scenes | | | | | | ETH3D | | | | | | | RealEstate10K | | | ScanNet | | |
| | Acc.↓ | | Comp.↓ | | NC.↑ | | Acc.↓ | | Comp.↓ | | NC.↑ | | RRA@5↑ | RTA@5↑ | AUC@30↑ | ATE↓ | RPE-T↓ | RPE-R↓ |
| | Mean | Med. | Mean | Med. | Mean | Med. | Mean | Med. | Mean | Med. | Mean | Med. | | | | | | |
| MVDUSt3R | 0.026 | 0.011 | 0.030 | 0.013 | 0.730 | 0.838 | 0.400 | 0.291 | 0.376 | 0.159 | 0.805 | 0.905 | 98.66 | 12.91 | 42.34 | 0.015 | 0.019 | 0.691 |
| VGGT | 0.020 | 0.008 | 0.029 | 0.016 | 0.694 | 0.790 | 0.263 | 0.188 | 0.197 | 0.120 | 0.855 | 0.961 | 99.51 | 15.75 | 50.06 | 0.015 | 0.015 | 0.500 |
| Hunyuan+MVDUSt3R | 0.027 | 0.012 | 0.032 | 0.012 | 0.699 | 0.806 | 0.405 | 0.288 | 0.399 | 0.166 | 0.802 | 0.887 | 98.36 | 12.40 | 41.97 | 0.016 | 0.019 | 0.668 |
| SVD+MVDUSt3R | 0.026 | 0.011 | 0.030 | 0.013 | 0.727 | 0.834 | 0.410 | 0.310 | 0.387 | 0.168 | 0.804 | 0.899 | 98.12 | 12.67 | 41.69 | 0.016 | 0.020 | 0.690 |
| CogVid+MVDUSt3R | 0.028 | 0.012 | 0.033 | 0.014 | 0.699 | 0.808 | 0.412 | 0.281 | 0.387 | 0.157 | 0.781 | 0.888 | 98.29 | 12.36 | 41.96 | 0.016 | 0.019 | 0.680 |
| Wan+MVDUSt3R | 0.027 | 0.011 | 0.032 | 0.012 | 0.712 | 0.825 | 0.401 | 0.297 | 0.386 | 0.164 | 0.797 | 0.910 | 98.28 | 12.30 | 42.12 | 0.016 | 0.019 | 0.680 |
| Wan+VGGT | 0.018 | 0.008 | 0.032 | 0.015 | 0.693 | 0.790 | 0.265 | 0.166 | 0.193 | 0.121 | 0.837 | 0.960 | 99.65 | 15.98 | 50.86 | 0.014 | 0.015 | 0.520 |

(a) log-MSE value in Eq. 2    (b) Acc.↓    (c) Comp.↓    (d) NC.↑

Figure 5: **MSE and pointmap quality on 7-Scenes vs. to stitching layer.** Lower MSE in the stitching layer correlates with better 3D reconstruction.

## 4.3 MAIN RESULTS: MODEL STITCHING

Stitching the 3D foundation models from Section 4.1 with a video VAE yields two variants: a VAE for Gaussian splats (AnySplat + video VAE) or a VAE capable of reconstructing pointmaps and camera poses (MVDust3R or VGGT + video VAE). In the following, we evaluate both variants.

**Evaluation protocol.** For 3DGS models, we evaluate novel-view synthesis on RealEstate10K (Zhou et al., 2018), with 8 source and 4 target images. For 3D reconstruction models, we follow Pi3 (Wang et al., 2026) and assess pointmap quality on 7Scenes (Shotton et al., 2013) and ETH3D (Schöps et al., 2017), and camera pose estimation on RealEstate10K and ScanNet (Dai et al., 2017). Specifically, Accuracy (Acc.), Completion (Comp.), and Normal Consistency (N.C.) are used for pointmap estimation, while camera pose estimation is evaluated with Relative Rotation Accuracy (RRA) and Relative Translation Accuracy (RTA) at 5° and their AUC up to 30°.

**Novel view synthesis.** Table 3 reports results on RealEstate10K. Stitching AnySplat onto any video model always improves over using AnySplat alone. We attribute the gains to the richer appearance representation of video VAE latents. The experiment is consistent with the results of Wonderland (Liang et al., 2025), where operating in latent space rather than RGB space also benefits 3DGS. Moreover, our stitched VAEs outperform the earlier VAE-based approaches. Remarkably, we surpass Prometheus3D and VideoRFSplat despite their use of camera poses and large-scale training data, showing that stitching high-performance 3D models is indeed an effective strategy to obtain powerful 3D VAEs.

**Pointmap reconstruction results.** Table 5 shows that stitching preserves the accuracy and completeness of the original 3D foundation models: both pointmap quality and camera pose accuracy barely change when using video encoder latents as input. The results confirm that stitching achieves its goal, to take advantage of the pretrained models' 3D reconstruction capabilities and repurpose them for generative modeling, without relying on large training datasets or labels.

## 4.4 ABLATIONS

**Effectiveness of MSE for finding stitching layer (Sec 3.1).** We pick the best layer for stitching according to a fairly simple criterion, namely the one that best supports a linear transfer of the encoder latents. To analyze the impact of this design, we train stitched decoders for the combination (Wan+VGGT) while varying the stitching index. In Fig. 5, we see that layers with lower stitching residual indeed yield better pointmaps, supporting the MSE of the linear stitching layer as our selection criterion.

This empirical trend is also consistent with existing theory: Theorem 1 in Insulla et al. (2025) shows that the stitching risk of the hybrid network, obtained by connecting the source model's early layers $f_1$ with the target model's latter layers $g_2$ via a linear map $S_{1,2}$, is upper-bounded by the MSE at the stitching layer,

$$\mathbb{E}\big[\|g_2(S_{1,2}f_1)(x) - g_2(f_2)(x)\|^2\big] \leq \kappa_2^2\,\mathbb{E}\big[\|S_{1,2}f_1(x) - f_2(x)\|^2\big], \tag{4}$$

where $\kappa_2$ is the Lipschitz constant of $g_2$. Thus, an MSE is related to the upper bound on the stitching error, supporting our use of MSE.

Furthermore, motivated by the observation of Insulla et al. (2025) that the right-hand side of Eq. 4 takes a similar form of kernel alignment, we investigate whether CKA (Kornblith et al., 2019) can track the trend of final performance. Figure 6 reports the CKA between the latent representation of the Wan VAE and the representations at different layer indices of VGGT (larger values indicate greater similarity). As shown, CKA captures the overall degradation in performance as the layer index increases; however, it is less precise than MSE in identifying the best layer, failing to capture that the best performance is achieved at layer 2. These results suggest that, in our setting, MSE is a more reliable indicator of transferability than CKA.

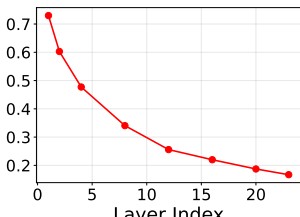

Figure 6: **CKA visualization.**

**Impact of direct reward finetuning (Sec 3.2).** As shown in Appendix D.1, direct reward finetuning is more effective than a pretrained video model on its own, as well as that same model finetuned on multi-view data, with each reward component contributing to the overall performance.

**Benefits of integrated vs. sequential 3D generation.** In Appendix D.2, we observe that an integrated approach is more robust to noise in the latent space, which suggests it may lead to more consistent 3D reconstruction from noise in the generation process.

Additional results in Appendix E demonstrate that VIST3A inherits prompt-based camera control from the video backbone (e.g., responding to instructions like "aerial droneshot"), that the stitching analysis generalizes across multiple VAE architectures, and that our finetuning does not degrade — and even slightly improves — video generation quality as measured by VBench.

**Additional results.** In Appendix E, we further show that VIST3A inherits prompt-based camera control from the video backbone, and the stitching layer analysis generalizes across multiple VAE architectures, and our finetuning does not degrade video generation quality on VBench.

## 5 CONCLUSION

We have presented VIST3A, a framework for training latent diffusion models that generate 3D content from text prompts. Our key idea is to employ model stitching as a way to integrate the generative abilities of modern video models with the 3D understanding of recent feedforward 3D models. We found that this strategy indeed leads to high-quality 3D VAEs, while not requiring labeled data or massive training runs. To then align a latent-space video generator with the stitched 3D decoder it feeds into, we design a reward-based finetuning strategy. Together, these two measures yield a family of text-to-3D models with high-quality, geometrically consistent 3D outputs. In passing, they extend 3D generation to other outputs of foundational 3D models, such as pointmaps and depthmaps. More broadly, we see great potential for model stitching as a general tool to combine two or more foundational neural networks, including latent generative models, into powerful end-to-end solutions.

## 6 ACKNOWLEDGMENTS

This work was supported by an unrestricted gift from Google, and by a grant from the Swiss National Supercomputing Centre within the Swiss AI Initiative (project a144). We thank Shengyu Huang for insightful discussions and feedback on the paper, particularly regarding the relationship to Representation Autoencoders. We are also grateful to Yongwei Chen for help with reviewing, validating, and cleaning the code.

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

## A  EXTENDED RELATED WORKS

**Pipeline-based 3D generation.**   A common design pattern chains multiple modules sequentially: a first stage generates multi-view images from text or a single image, and a second stage lifts those views into a 3D representation (Tang et al., 2024a; Xu et al., 2024b; Zhang et al., 2024b; Li et al., 2024a; Park et al., 2025). The reconstruction step is handled either by a large feedforward model such as LRM (Hong et al., 2024), or by per-scene optimization of NeRFs (Mildenhall et al., 2021) or 3DGS (Kerbl et al., 2023) (Liu et al., 2024; Yu et al., 2025b; Gao et al., 2024; Sun et al., 2025; Wang et al., 2025a). In both cases, the generative and reconstruction stages are trained independently, making the overall system prone to error accumulation (e.g., view inconsistency, texture flicker) and less robust to latent-space perturbations than joint formulations in latent space (See Section E). The per-scene optimization variant additionally incurs substantial inference cost.

A separate line of work pursues progressive expansion and refinement (Yu et al., 2024; Ni et al., 2025; Chen et al., 2025; Fridman et al., 2023; Feng et al., 2025; Yu et al., 2025a). Some methods adopt iterative warping and inpainting (Yu et al., 2024; Ni et al., 2025; Fridman et al., 2023; Yu et al., 2025a), while others leverage video generative models to unfold 3D scenes progressively (Chen et al., 2025; Feng et al., 2025). Beyond these, additional works propose elaborate multi-stage pipelines that further increase complexity (Yang et al., 2025b; Ost et al., 2025). However, such designs are overly complex and suffer from slow inference.

**Alignment for text-to-2D models.**   Several strategies have been proposed to align pretrained text-to-image models with human preferences: direct finetuning with scalar rewards (Clark et al., 2024; Xu et al., 2023; Prabhudesai et al., 2024; Wu et al., 2024c; Shen et al., 2025), Reward Weighted Regression (Peng et al., 2019; Lee et al., 2023), Direct Preference Optimization (Rafailov et al., 2023; Yang et al., 2024), and PPO-based policy gradients (Black et al., 2024; Fan et al., 2023; Liu et al., 2025). Among these, direct reward finetuning offers gradient-based feedback that propagates through the full denoising trajectory, which makes it a natural fit for our setting, where the reward depends on the decoded 3D output.

**Concurrent works on interchanging parts of VAEs.**   Concurrently, Chen et al. (2026) replace pretrained VAE encoders with visual encoders that extract semantic representations. Their approach does not explicitly account for the compatibility between the substituted encoder's representation space and the downstream decoder, and consequently requires extensive retraining. In our framework, we measure the linear transferability between the VAE latent space and each layer of the 3D model, and stitch at the most compatible layer, which yields a well-initialized decoder that needs only lightweight finetuning.

## B  METHODOLOGY DETAILS AND IMPLEMENTATION

### B.1  MODEL STITCHING

**Stitching layer.**   We implement the stitching layer $\mathbf{S}$ as a single Conv3D layer. Relying only on Conv3D parameters to align the spatial and temporal dimensions between the latent and the features from $F_{k+1:l}$ can result in unnatural configurations, such as excessively large padding size. To address this, we first interpolate the latent representation to the target dimensions and then apply Conv3D, which provides a cleaner alignment of spatial and temporal dimensions. This design still admits a closed-form expression of the stitching layer, as shown in Eq. 2.

**Loss function for each 3D model.**   We train the stitched VAE using an $\ell_1$ loss between its outputs and those of the original 3D model. Since 3D model outputs often consist of multiple components, we compute the $\ell_1$ loss for each component separately and then aggregate them with a weighted sum. Assigning equal weights can destabilize training and even cause divergence, since some components (e.g., confidence terms) have much larger scales than others. To mitigate this, we reweight the component losses to approximately balance their scales. The specific weighting strategy is adapted to each 3D model as follows:

- **MVDUSt3R.** The outputs consist of pointmaps, confidence scores for the pointmaps, and 3D Gaussian primitives We assign a weight of $10^{-2}$ to the confidence term, while pointmap and Gaussian primitive losses are left unscaled.

- **VGGT.** Outputs include pointmaps, depth maps, camera poses, and confidence for both pointmaps and depth. In addition, following VGGT's practice, we add gradient-based regularization losses on pointmaps and depth maps. We weight the pose loss by 5, all confidence terms by $5 \times 10^{-3}$, and gradient regularization losses by $5 \times 10^{-3}$. Other losses remain unscaled.

- **AnySplat.** Outputs include depth maps, Gaussian primitives, confidence for both depth and Gaussian primitives, camera poses, and anchor features. Additionally, we introduce gradient-based regularization losses on the depth maps. We weight all confidence terms by $10^{-2}$, gradient regularization losses by $5 \times 10^{-3}$, Gaussian scale parameters by 10, and anchor features by 0.1. Depth and other Gaussian parameters are left unscaled.

**Hyperparameters and implementation details.** For the stitching layer **S**, we adopt a single 3D convolution with kernel size, stride, and padding chosen to align the latent features from the video VAE with the representation space of each 3D model:

- **MVDUSt3R:** a 3D convolution with kernel size $5 \times 7 \times 7$, output channels 1024, stride $1 \times 3 \times 3$, and padding $2 \times 0 \times 0$.

- **VGGT:** a 3D convolution with kernel size $5 \times 3 \times 3$, output channels 1024, stride $1 \times 2 \times 2$, and padding $2 \times 1 \times 1$.

- **AnySplat:** a 3D convolution with kernel size $5 \times 3 \times 3$, output channels 1024, stride $1 \times 2 \times 2$, and padding $2 \times 1 \times 1$.

Before applying the convolution, the interpolation layer recovers the temporal dimension compressed by the video VAE and adjusts the spatial size so that it matches the resolution expected by the feedforward 3D model. The input resolution of the video VAE is set to $384 \times 384$ for MV-DUSt3R and $512 \times 512$ for both AnySplat and VGGT, as these configurations empirically yield stable training for the respective generative backbones. We employ LoRA with rank $r = 64$ and scaling factor $a = 32$ to Conv2D and linear layers across all cases.

### B.2 DIRECT REWARD FINETUNING

**Reward details.** We combine CLIP-based scores and HPSv2.1 human preference scores to construct rewards for both multi-view image quality and 3D representation quality. Specifically, we use DFN (Fang et al., 2024) as the CLIP model and HPSv2.1 (Wu et al., 2023). Given an image $I$ and its associated prompt $c$, we denote the HPSv2.1 score as $s_{\text{hps}}$ and the DFN CLIP score as $s_{\text{clip}}$. The quality reward is then defined as

$$R_{\text{quality}}(I, c) = s_{\text{clip}}(I, c) + s_{\text{hps}}(I, c) - 2, \tag{5}$$

which implies that maximizing the reward is equivalent to maximizing the underlying score.

For the **multi-view image quality reward**, we compute the scores using the multi-view images decoded from the video decoder and their corresponding prompts. For the **3D representation quality reward**, we compute the scores using the rendered images obtained from the 3D representation reconstructed by the stitched decoder, together with the same prompts.

The 3D consistency reward is computed as a combination of the pixel-level $\ell_1$ loss and the LPIPS between a decoded multi-view image and its corresponding rendering from the reconstructed 3D representation. Formally, given a decoded image $I_{\text{decode}}$ and the estimated camera pose $\hat{\pi}$ predicted by the stitched decoder, we obtain the rendered image $I_{\text{rendered}}(\hat{\pi})$ from the 3D representation. The consistency reward is then defined as

$$R_{\text{consistency}}(I_{\text{decode}}, I_{\text{rendered}}(\hat{\pi})) = -|I_{\text{decode}} - I_{\text{rendered}}(\hat{\pi})|_1 - 0.25 \times \text{LPIPS}\left(I_{\text{decode}}, I_{\text{rendered}}(\hat{\pi})\right). \tag{6}$$

Here, the negative sign ensures that maximizing the reward corresponds to minimizing both the $\ell_1$ distance and the perceptual discrepancy between the decoded and rendered images.

However, applying these rewards to all decoded multi-view images and their rendered counterparts is computationally expensive. To reduce computational cost, we compute all rewards only on two sampled decoded views and their corresponding rendered images. The final reward is then obtained by a weighted combination of the three components: the multi-view image quality reward and the 3D representation quality reward are each scaled by $1/16$, while the 3D consistency reward is scaled by 0.05. These scaled terms are summed to form the overall training reward.

**Alignment Algorithm.** For alignment, we adopt DRTune (Wu et al., 2024c)-style direct reward finetuning, which enables stable reward optimization through selective gradient computation. We outline one training iteration of our finetuning in Algorithm 1. First, we calculate the generative loss using multi-view datasets, then simulate the denoising process. Since matching the full number of inference-time denoising steps during training is costly, we instead sample $t$ steps from a reduced range $[T_1, T_2]$ to lower the computational burden. Additionally, to reduce time and memory costs, we only enable gradient computation at $K$ selected training steps $t_{\text{train}}$ out of the total $t$ steps. Following DRTune, the input $z_\tau$ to the generative model is detached at each step to stabilize optimization. Finally, we calculate the reward from the sampled latent and combine it with the generative loss by subtraction (for maximization) before backpropagation and parameter updates.

---

**Algorithm 1** One Training Iteration of Alignment Training

1: **Input:** generative model $\theta$, reward $r$, sampling step range $[T_1, T_2]$, # of gradient enabled steps $K$, prompt $c$, data $D$.
2: $L_{\text{gen}} \leftarrow$ calculate generative loss with $D$
3: $t \sim \text{Uniform}(T_1, T_2)$ ▷ Sample number of denoising steps
4: $z_T \sim \mathcal{N}(0, I)$ ▷ Initialize starting noise
5: Define $t$-step schedule $\{\tau_j\}_{j=0}^t$ with $\tau_0 = T, \tau_t = 0$
6: $t_{\text{train}} \leftarrow$ randomly select $K$ indices from $\{1, \ldots, t\}$
7: **for** $j = 1$ to $t$ **do** ▷ Denoising from $T$ to $0$
8:     $\hat{z}_{\tau_j} \leftarrow \texttt{stop\_grad}(z_{\tau_j})$
9:     **if** $j \in t_{\text{train}}$ **then**
10:         prediction $\leftarrow \text{model}(\theta, \hat{z}_{\tau_j}, \tau_j)$
11:     **else**
12:         **no\_grad:** prediction $\leftarrow \text{model}(\theta, \hat{z}_{\tau_j}, \tau_j)$
13:     $z_{\tau_{j+1}} \leftarrow \text{update}(z_{\tau_{j-1}}, \text{prediction})$
14: $r(z_0, c) \leftarrow$ Calculate reward of generated latent.
15: $L_{\text{total}} \leftarrow L_{\text{gen}} - r(z_0, c)$
16: Backpropagate $\nabla_\theta L_{\text{total}}$, then optimize $\theta$

---

**Hyperparameter in sampling process.** For generating samples required in the $[T_1, T_2]$ direct reward tuning stage, we set $T_1 = 10$ and $T_2 = 50$ in Algorithm 1, ensuring that the number of diffusion steps is smaller than the typical steps in inference. The number of gradient-enabled steps is set to $K = 2$ to reduce memory consumption during training. For scheduling, we adopt the default scheduler from Wan 2.1 (Wan et al., 2025).

## C DETAILS ON EXPERIMENTAL SETUPS

### C.1 TRAINING SETUP

**Setup for stitching layer search.** To identify the stitching layer, we rely on representations from the feedforward 3D model and the corresponding latents computed on the same dataset. Specifically, we utilize a subset of the DL3DV dataset, comprising 200 scenes for VGGT, 800 scenes for AnySplat, and 3,200 scenes for MVDUSt3R, with only 13 views per scene used for the search. We limit our search to the encoder layers of each model, as we observe that MSE values consistently increase within deeper layer indices.

**Setup for stitched VAE finetuning.** We train on a combination of the DL3DV and ScanNet datasets, defining one epoch as a full pass over DL3DV and two passes over ScanNet. For each training iteration, a number of scenes are sampled according to the batch size. From each selected scene, we randomly sample 9 or 13 views to serve as input samples for training. The models are trained for 50 epochs in total. The batch sizes are set to 12 for VGGT, 24 for MVDUSt3R, and 12 for AnySplat. The learning rate is fixed at $2 \times 10^{-4}$ for all models with cosine decay scheduling and 500-step warmup. For training, we use AdamW (Loshchilov & Hutter, 2019), apply gradient clipping with a norm threshold of 1.0, and use gradient checkpointing on each stitched VAE block to reduce memory consumption. In addition to LoRA parameters, for AnySplat and VGGT, we also finetune register tokens and class tokens. This is necessary because we remove the earlier layers that originally process these tokens into intermediate representations, requiring adaptation of the token handling mechanism. We further utilize gradient checkpointing for every stitched VAE block.

**Setup for generative model finetuning.** We finetune the generative models using only the DL3DV dataset. For generative loss computation, we use a batch size of 12 with 13 views per scene. Reward calculation uses a prompt batch size of 4, with 13 views for AnySplat and MVDUSt3R, and 9 views for VGGT. We again adopt AdamW with a learning rate of $1 \times 10^{-4}$, apply gradient clipping at a 0.1 norm, and train LoRA parameters with rank 8 and alpha 16. Gradient checkpointing is enabled for all model blocks to reduce memory usage.

## C.2 DETAILED EVALUATION PROTOCOL

**Details for 3D generation evaluation.** For T3Bench, we evaluate on all 300 prompts, in contrast to prior works that considered only the 100 single-object-with-surroundings subset. SceneBench is evaluated on 80 prompts from the Prometheus3D (Yang et al., 2025c) prompt set, targeting scene-level generation. For DPG-Bench, we sample 100 prompts from the original 1K-prompt dataset.

For Matrix3D-omni, we used their official code for text-to-generation and employed Panorama LRM for reconstruction during inference. For SDS-based methods like SplatFlow and Director3D that perform refinement, we evaluated the final results after SDS optimization. We generate 13 frames for all models using 80 denoising steps, and apply classifier-free guidance (Ho & Salimans, 2021) with a scale of 7.5. We observed that the Gaussian splatting produced by the MVDUSt3R model does not generalize well across diverse domains, often failing to estimate the scale of primitives. To address this issue, we refined the Gaussian primitives using the source view for 100 optimization steps, minimizing a reconstruction loss defined as MSE + 0.05 × LPIPS. For this refinement, we used the Adam optimizer with separate learning rates for each parameter group: 2e-4 for means, 5e-4 for opacity, 5e-4 for scale, 1e-4 for rotation, and 0 for rgbs. This lightweight refinement effectively corrected the scale estimation errors. For our text-to-3DGS evaluation, we render 8 random viewpoints from the generated Gaussian Splatting representations for assessment.

We evaluate our method and baselines across a range of metrics. To measure the semantic similarity between the input prompt and the rendered images of the generated 3DGS, we compute the CLIP score using the clip-vit-base-patch16 model. Additionally, we adopt the VBench (Huang et al., 2024) framework to assess key image properties. For Imaging Quality, which targets low-level distortions, we employ the same MUSIQ model (Ke et al., 2021) in VBench. For Aesthetic Quality, we use the LAION aesthetic predictor to evaluate the color richness and artistic merits, again following VBench. The predictor's native 0-10 rating is linearly normalized to a 0-1 scale for our analysis.

For a more comprehensive assessment of generative quality, we utilize the Unified Reward model (Wang et al., 2025d), which is based on the powerful Qwen 2.5-7B Vision Language Model (Team, 2025)[2]. This provides fine-grained, pointwise scores on complex attributes equipped with a powerful understanding capability. By feeding the input prompt and rendered images into a format adapted from the official implementation script[3], we obtain scores for three key aspects:

- *Alignment*: How well the image content matches the text prompt.
- *Coherence*: The logical and visual consistency of the image, free of distortions.
- *Style*: The aesthetic appeal of the image, independent of prompt accuracy.

This suite of metrics enables a robust and multifaceted evaluation of our model's performance.

**Details for model stitching evaluation.** For novel-view synthesis, we follow prior works (Go et al., 2025a;b) and adopt an 8-frame input setup to evaluate performance on 4 target views. To accommodate the fixed-length input requirements of video VAE architectures due to temporal compression, we pad shorter sequences by duplicating the final frame. For estimating the camera poses of the target views, we adopt the strategy from AnySplat (Jiang et al., 2025), which jointly predicts the poses and renders the corresponding images. This contrasts with previous VAE-based methods that presume access to ground-truth camera poses for rendering.

For pointmap and camera pose estimation evaluation, we use a 13-frame input setup. Since our stitched VAE's encoder is a video VAE, we arrange the multi-view images (typically provided un-ordered by previous works) into sequences with smooth view transitions to resemble video input. We adopt Pi3 (Wang et al., 2026) official evaluation code and follow their preprocessing pipeline.

## C.3 SELECTION CRITERION FOR 3D MODELS

In this section, we elaborate on the criteria used for selecting 3D foundation models in VIST3A.

---

[2] https://huggingface.co/CodeGoat24/UnifiedReward-qwen-7b
[3] https://github.com/CodeGoat24/UnifiedReward/blob/main/inference_qwen/image_generation/qwen_point_score_ACS_image_generation.py

Table 6: **Ablation study on direct reward finetuning on SceneBench.** We compare (1) no finetuning; (2) multi-view-only finetuning (generative loss only); (3) reward tuning with 3D-consistency reward only; (4) reward tuning with quality reward only; and (5) reward tuning with both rewards (full).

| Method | Imaging | Aesthetic | CLIP | Unified Reward | | |
| --- | --- | --- | --- | --- | --- | --- |
| | | | | Alignment | Coherent | Style |
| Finetuning-free | 50.56 | 53.70 | 28.14 | 3.101 | 3.354 | 3.393 |
| Multi-view only | 54.56 | 52.08 | 29.71 | 3.622 | 3.834 | 3.351 |
| Multi-view + 3D Consistency | 38.67 | 50.59 | 29.77 | 3.581 | 3.767 | 3.275 |
| Multi-view + Quality | 62.27 | **58.23** | **30.34** | 3.643 | 3.842 | 3.358 |
| Ours | **64.87** | 56.96 | 30.18 | **3.667** | **3.862** | **3.400** |

Our primary criterion for selecting a 3D backbone is the scale of the pretraining dataset, which mainly determines the generalizability of the model. This is because the coupled 3D model should cover highly diverse domains of video generative models. AnySplat (Jiang et al., 2025) is selected as it represents one of the few Gaussian Splatting models trained on such a scale. For pointmap-based models, we utilize MVDUSt3R (Wang et al., 2024) for its balance of dataset scale and efficiency (facilitating feasibility checks), and VGGT (Wang et al., 2025b) for its superior performance and pretraining scale.

# D    FURTHER ABLATION STUDIES

## D.1    IMPACT OF DIRECT REWARD FINETUNING

In the following, we conduct an ablation study to analyze the effects of our direct reward finetuning, comparing our full method against four well-defined baselines:

- (1) Finetuning-free: Here, we use the original pretrained video model. Since our finetuning freezes the encoder, its latent space remains compatible with our 3D stitched decoder.
- (2) Multi-view Only: The model finetuned with only the flow-matching loss on multi-view data, serving as our primary baseline before rewards are introduced.
- (3) Multi-view + Consistency: The model finetuned with both the multi-view loss and the 3D-consistency reward. This isolates the impact on the 3D consistency reward.
- (4) Multi-view + Quality: The model finetuned with both the multi-view loss and the quality reward. This isolates its impact on quality reward.

To ensure a fair comparison against reward-based methods, which often take more time for one training iteration, the finetuning variant on multi-view data was trained for the same wall-clock duration.

Table 6 reports the quantitative results. The finetuning-free baseline yields the lowest performance. Lacking any 3D-aware training, it frequently produces geometrically inconsistent outputs and suffers from significant visual artifacts when its native resolution is adapted to our 3D decoder. Introducing multi-view supervision (Multi-view Only) substantially improves 3D consistency and overall performance, confirming the value of this training signal.

The reward components have distinct effects when added to the multi-view objective. Training with the 3D-consistency reward (Multi-view + Consistency) leads to a notable performance drop, as the model optimizes for geometric correctness at the expense of detail, resulting in overly blurred images. Conversely, adding the quality reward (Multi-view + Quality) achieves substantial improvements across most metrics by enhancing prompt coherence and aesthetic appeal.

Finally, our full method, which combines both rewards with multi-view training, achieves the best imaging quality and Unified Reward scores. While its aesthetic and CLIP scores are slightly below the Multi-view + Quality variant, the marked improvement in imaging quality demonstrates that our combined objective successfully guides the model to generate visually sharp and geometrically faithful 3D representations.

Rendered 3DGS | Decoded Frames

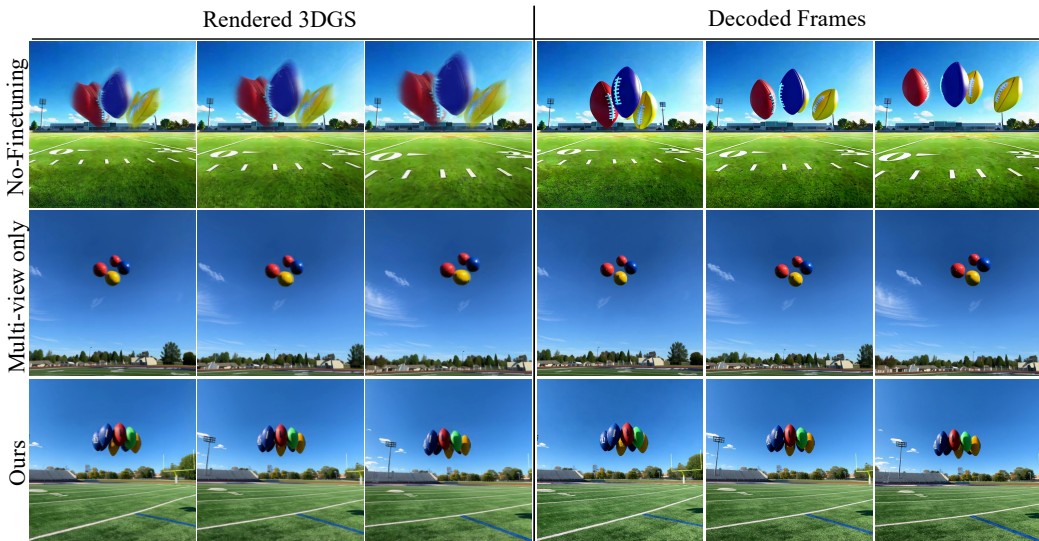

*"In the open expanse of a school's sports field, under the clear blue sky of a radiant sunny day, four vibrant American footballs are captured in mid-flight. The footballs, featuring hues of red, blue, yellow, and green, are spherical in shape, contrasting sharply with the green turf below. Each ball glistens in the sunlight as they arc gracefully above the field, momentarily suspended against the backdrop of a few wispy clouds."*

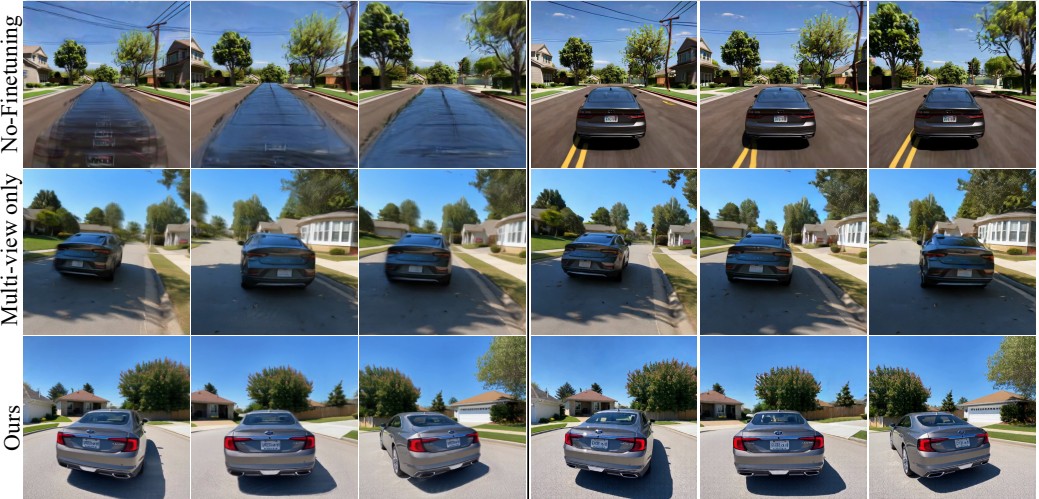

*A car parked on a street, its rear end facing away from the viewer against a house and trees backdrop.*

Figure 7: **Qualitative comparison of different finetuning strategies.** Pretrained video model (No-Finetuning) produces dynamic videos, causing severe ghosting in the reconstructed 3D scenes. Multi-view finetuning (Multi-view only) reduces this motion and improves 3D consistency but introduces semantic and quality degradations, while our direct reward tuning (Ours) yields sharper renderings that better align with the input text prompts.

To further analyze the effect of our training strategy, we provide a qualitative comparison between the pretrained video model, the multi-view-only finetuning baseline, and our reward-tuned model in Fig. 7. As shown in the decoded frames, the **No-Finetuning** baseline (pretrained video generator) often produces dynamic videos with temporal motion. This leads to severe ghosting artifacts in the generated 3D scenes, clearly visible as multiple outlines of the footballs and the car.

The **Multi-view only** baseline (second row) effectively suppresses this motion, enforcing 3D consistency. However, relying solely on a multi-view dataset limits the model's generalizability, and the lack of alignment with the decoder hinders the model from generating latents that are well reconstructible by the decoder. As a result, the generations exhibit semantic and quality degradations: the "green" football is missing in the first example, and the car becomes blurrier in the second example.

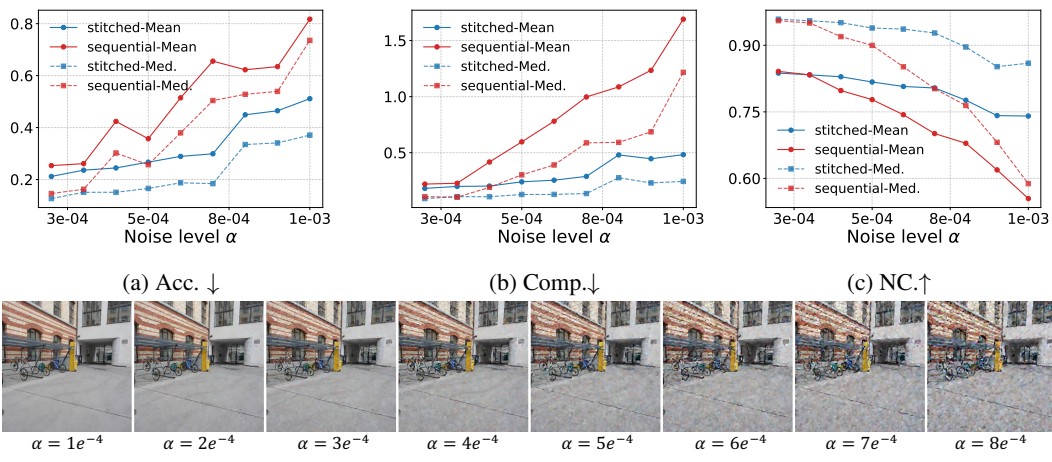

(a) Acc. ↓        (b) Comp.↓        (c) NC.↑

$\alpha = 1e^{-4}$   $\alpha = 2e^{-4}$   $\alpha = 3e^{-4}$   $\alpha = 4e^{-4}$   $\alpha = 5e^{-4}$   $\alpha = 6e^{-4}$   $\alpha = 7e^{-4}$   $\alpha = 8e^{-4}$

(d) Reconstructed images through VAE according to $\alpha$

Figure 8: **Pointmap estimation performance comparison on ETH3D dataset between the stitched VGGT and the sequential approach (VAE followed by VGGT) under varying noise scales injected into the latent space.** The stitched model demonstrates greater robustness to noise injection in the VAE.

In contrast, our direct reward tuning (**Ours**) applies rewards to the decoded 3D representation of the stitched decoder, explicitly encouraging high reconstruction quality, 3D consistency, and better alignment with the text prompt. Consequently, it produces sharper and more faithful renderings than the baselines in both examples.

## D.2   BENEFITS OF INTEGRATED VS. SEQUENTIAL 3D GENERATION

In our model-stitching design, generation and reconstruction take place in the shared latent space of the video diffusion VAE and the stitched 3D decoder. A common alternative is a sequential pipeline that decodes latents into RGB frames before applying a feedforward 3D model (e.g., VGGT) without further adaptation. To probe the core benefit of our unified formulation, we injected controlled perturbations into the latent representation, using

$$z' = z + \alpha \|z\| \epsilon, \quad \epsilon \sim \mathcal{N}(0, I), \tag{7}$$

where $\alpha$ is a scalar controlling the perturbation strength. We then compared two paths: (i) decode the corrupted latent to RGB and feed the images sequentially into the original VGGT (baseline), and (ii) directly input the noised latent into our stitched 3D decoder (unified latent framework).

Figure 8 reports pointmap estimation performance on ETH3D as a function of noise level $\alpha$. Our stitched VGGT consistently outperforms the sequential decode-and-reconstruct pipeline under noise injection, indicating that the VAE decoder in the sequential path amplifies errors. Moreover, as shown in Fig. 8d, the performance gap is observed even at noise levels ($\alpha = 1e^{-4}$ to $2e^{-4}$) where visual artifacts are hardly perceptible. This suggests that the unified design offers stronger robustness, as imperceptible perturbations from the noise of generative processes can already degrade the sequential pipeline.

## E   ADDITIONAL RESULTS

To comprehensively validate each component of **VIST3A**, we present additional experiments in this section.

**Prompt-based camera control.** Modern video generative model, including Wan 2.1, can reflect camera-related prompts such as "aerial view" and "camera pans left to right" in the generated videos. As our framework is built upon the Wan 2.1 backbone, we inherit this property, enabling text-driven viewpoint control in 3D generation. As illustrated in Fig. 9, the prompt containing "Aerial dronshot" induces a high-angle, downward-looking perspective (Top), whereas the prompt with "Camera pans left to right" results in a horizontal sweeping motion that traverses the scene from left

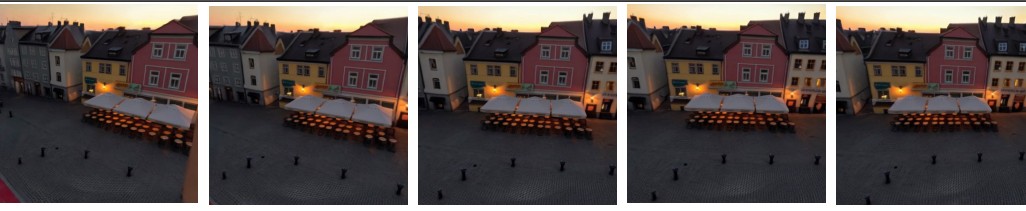

*"**Aerial dronshot** of a cozy outdoor café on a cobblestone street in a European town at sunset."*

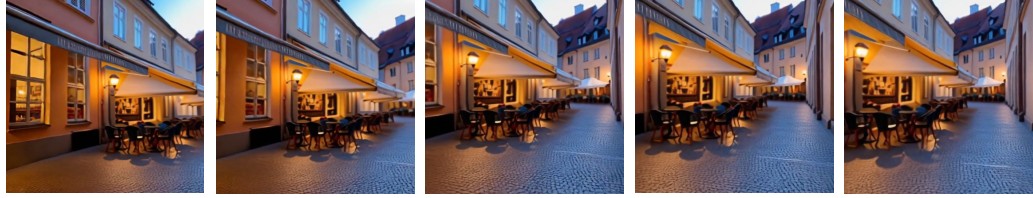

*"**Camera pans left to right** around a cozy outdoor café on a cobblestone street in a European town at sunset."*

Figure 9: **Viewpoint control in 3DGS generation with prompts. (Top)** The prompt containing "Aerial dronshot" results in a high-angle downward perspective. **(Bottom)** The prompt containing "Camera pans left to right" generates a trajectory where the viewpoint sweeps across the scene from left to right.

Table 7: **Video generation performance comparison between the original video generator and VIST3A on VBench.**

| Model | Quality Score | Semantic Score | Total Score |
|---|---|---|---|
| Original video generator | 0.7827 | **0.7275** | 0.7716 |
| VIST3A (Wan + AnySplat) | **0.8143** | 0.7143 | **0.7943** |

to right (Bottom). This shows that our model effectively inherits the semantic camera instruction understanding capability of pretrained video generative models.

**Impact of VIST3A finetuning on video generation performance.** We aim for video generative models to produce 3D-consistent frames, and we propose a finetuning strategy toward this goal. Here, we evaluate how much this finetuning degrades the original model's generative capability. To measure this, we use VBench (Huang et al., 2024), a well-established benchmark for video generation. We generate videos using the original model by matching the resolution to the VIST3A inference setting, and compare the video results of the finetuned model with VIST3A with Wan and AnySplat.

As shown in Table 7, the VIST3A-adapted model achieves a higher quality score and a higher total score than the original video generator, with only a marginal difference in the Semantic Score. This demonstrates that VIST3A's finetuning does not significantly degrade the video model's generative capability—in fact, the overall VBench performance is slightly improved under the VIST3A setup.

**Analysis on searched stitching index.** In Section 4.4, we showed that earlier layers in the network tend to be more linearly correspondent. We extend this analysis to various VAE architectures, including CogVideoX, SVD, Hunyuan, and Wan, paired with MVDUSt3R and AnySplat, to observe the generalizability of this finding.

Figure 10 shows the log-MSE values measuring linear transferability between latents and the feedforward 3D model's representations. From the results, early layers of 3d models consistently show lower MSE values across all VAE-feedforward 3D model combinations. This supports the hypothesis that latent representations capture low-level features for input reconstruction, which are more linearly transferable to the early layers of the feedforward 3D model that also encode such features. However, the results reveal an important distinction: while relative MSE ordering within each VAE architecture correlates with stitching performance (as in Section 4.4), absolute MSE values across different VAEs do not predict cross-architecture performance. For instance, CogVideoX + AnySplat achieves the lowest absolute MSE (0.008) but delivers 21.32 PSNR in Table 3, while SVD + AnyS-

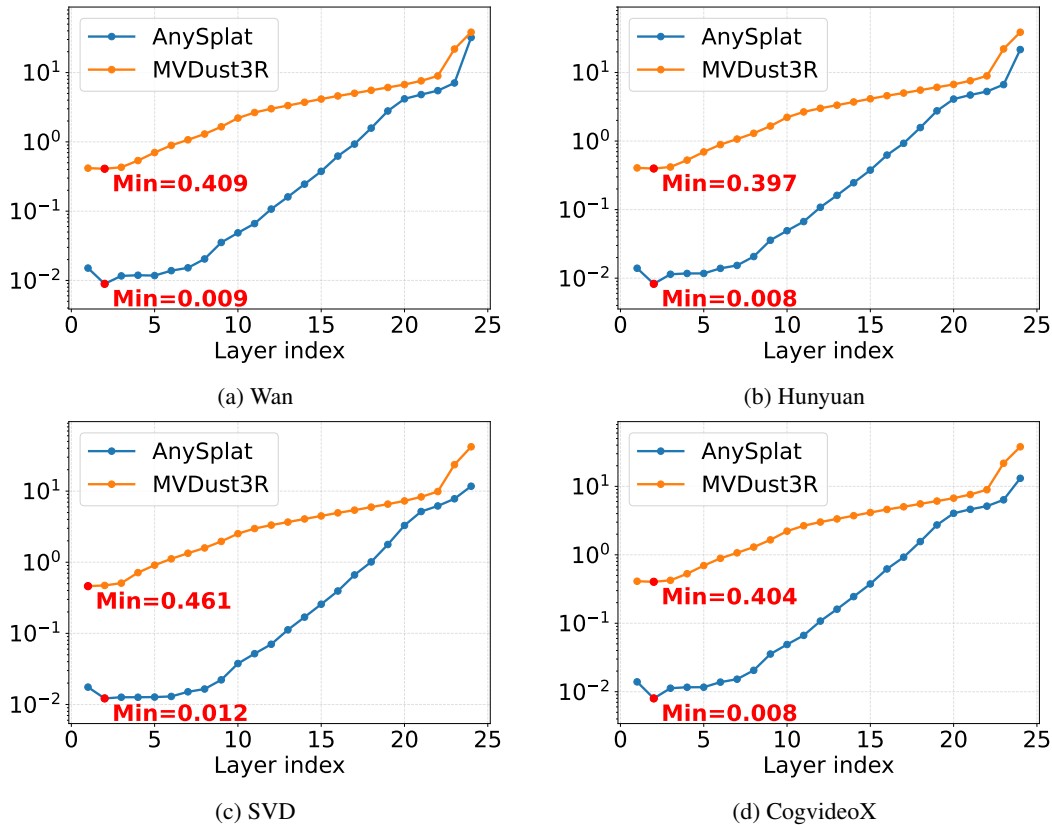

Figure 10: **Log-MSE values in Eq. 2 across various video VAEs.** Early layers of feedforward models show lower MSE values within each VAE architecture. While lower MSE correlates with better stitching performance within the same VAE (e.g., layer 2 outperforms layer 16 for Wan in Fig 5), absolute MSE values cannot predict performance across different VAE architectures. For instance, despite CogVideoX and Hunyuan + AnySplat having the lowest absolute MSE (0.008), SVD + AnySplat achieves the best performance (21.48 PSNR) in Table 3.

plat with a higher MSE (0.012) achieves superior performance at 21.48 PSNR. This indicates that optimal stitching layers must be identified independently for each VAE-3D model pair.

**Additional qualitative results.** We present additional qualitative results of VIST3A with Wan + AnySplat in Fig. 11–13. Text-to-pointmap generation results obtained by combining VGGT with Wan through VIST3A are shown in Fig. 14. Finally, Fig. 15 illustrates VIST3A results with MV-Dust3R + Wan.

## F    LIMITATIONS

While our approach demonstrates strong results, it also has certain limitations. Our stitched model inherits its encoder from a video generation model, which is inherently designed for sequential, temporally coherent video input. Consequently, its performance is not guaranteed for arbitrarily unordered inputs, such as typical multi-view image datasets. To ensure the encoder operates effectively, the input images must be arranged into a coherent sequence that simulates the smooth view transitions of a video clip.

## G    USE OF LARGE LANGUAGE MODELS

LLMs were used exclusively for text polishing and grammar refinement.

| Director3D | Prometheus3D | SplatFlow | VideoRFSplat | VIST3A: Wan + AnySplat |
|---|---|---|---|---|

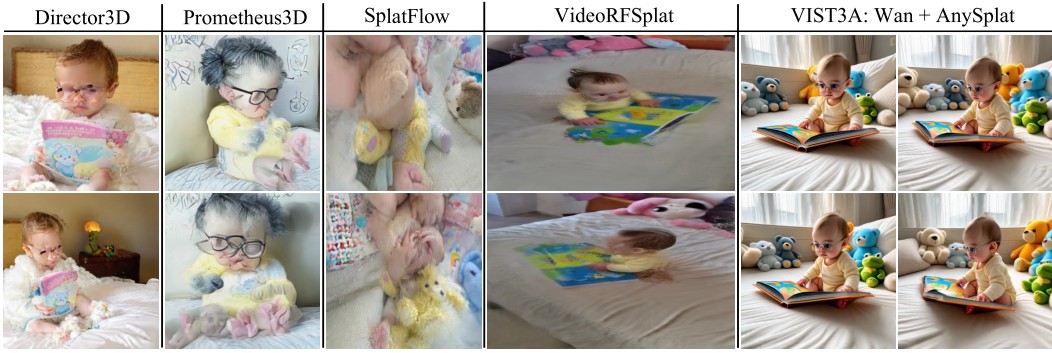

*"A small infant with round, silver-framed glasses perched on their nose is comfortably sitting in the center of a plush white bed. The child, dressed in a pale yellow onesie, holds an open, colorful picture book with both tiny hands, appearing to gaze intently at the illustrations. Surrounding the infant are an assortment of plush toys, including a fluffy blue bear and a soft green frog, scattered about the soft, cream-colored bedspread."*

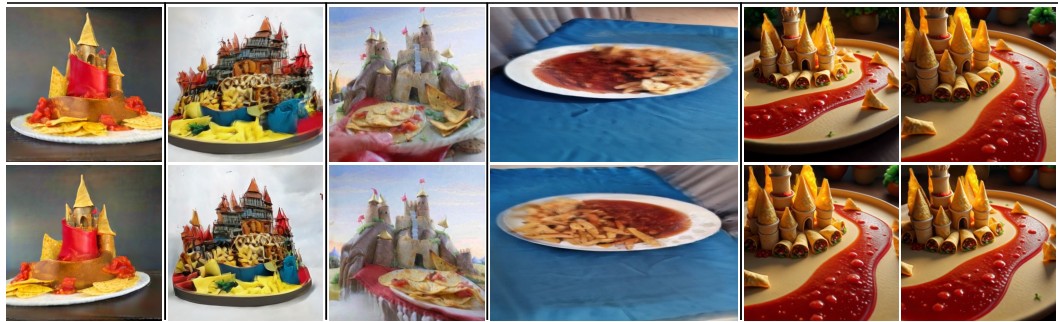

*"An imaginative scene unfolds with a castle intricately constructed from golden tortilla chips, its towers and walls standing tall amidst a flowing river of vibrant red salsa. Surrounding the edible fortress, tiny burritos, wrapped in soft tortillas with visible fillings, appear to be animated and meandering along the banks of the salsa river. The entire whimsical landscape is set upon a large plate, suggesting a playful, culinary creation."*

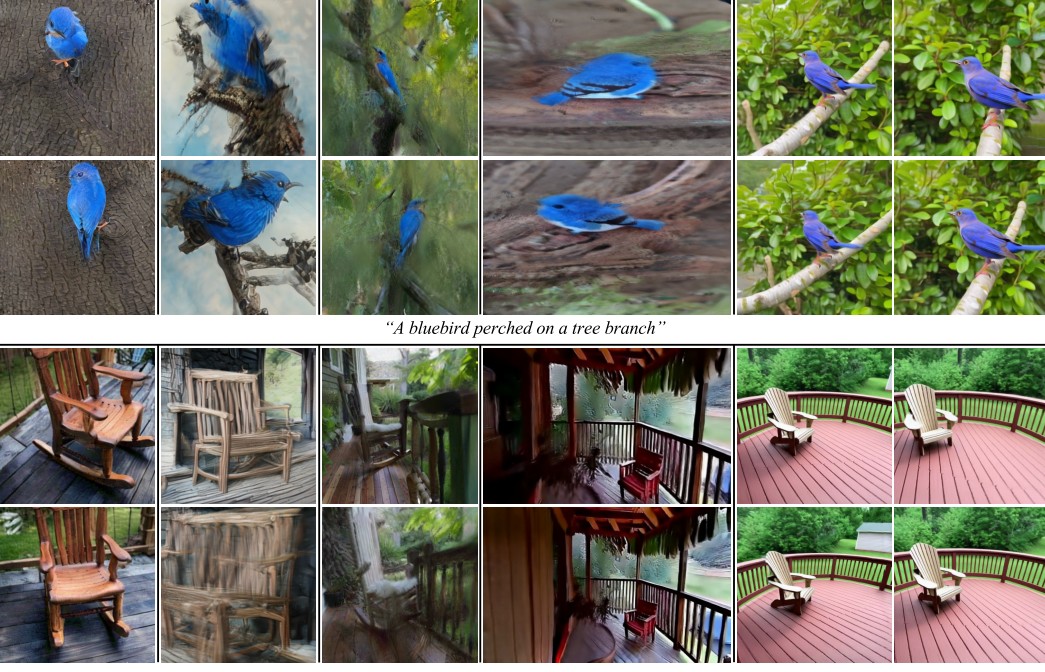

*"A bluebird perched on a tree branch"*

*"A wooden rocking chair on a porch"*

Figure 11: **Qualitative comparison of 3DGS generation.** The top two rows show samples from DPG-Bench, and the bottom two rows present samples from T3Bench. VIST3A generates realistic scenes with fine-grained details that faithfully reflect the input prompt, outperforming baselines.

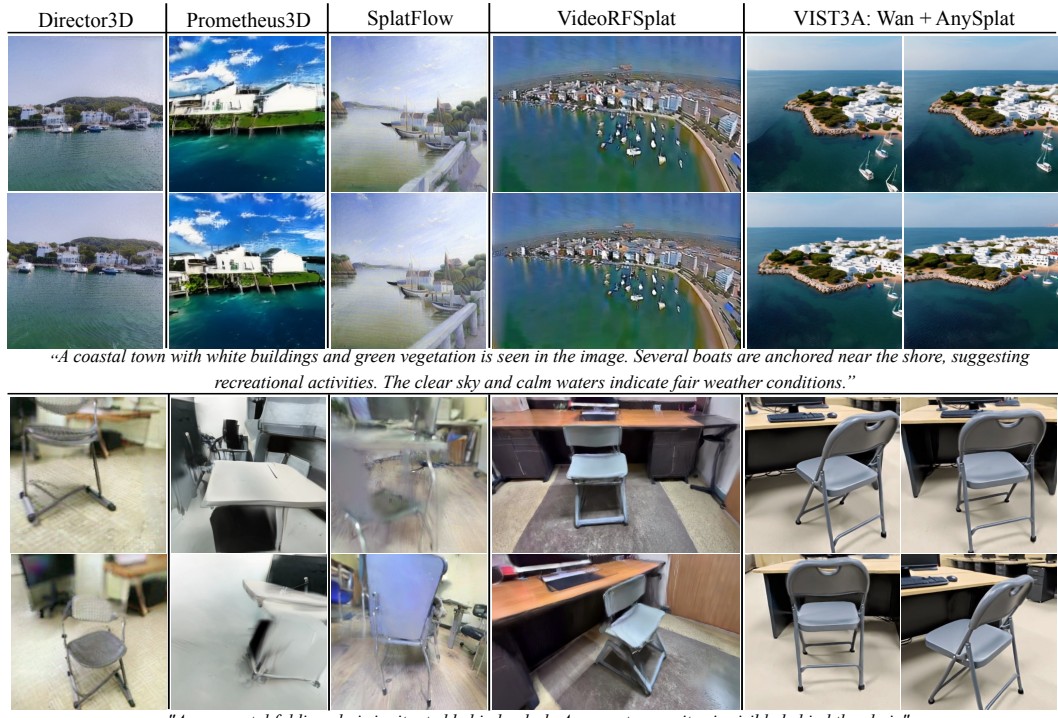

Figure 12: **Qualitative comparison of 3DGS generation on SceneBench.** VIST3A outperforms baselines by generating higher-fidelity scenes with accurate geometry and appearance.

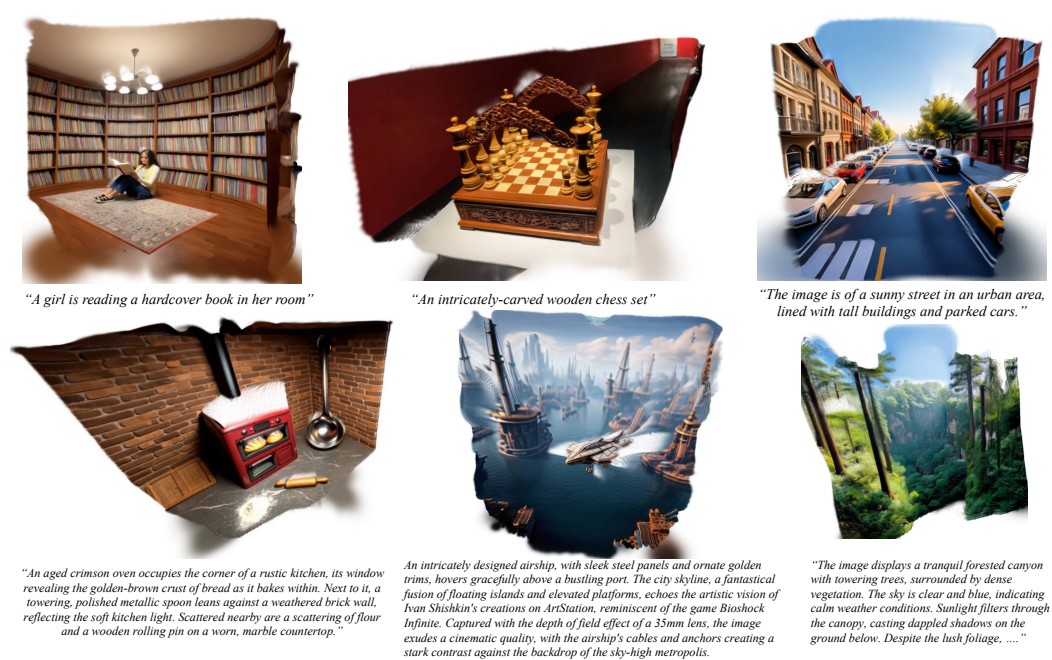

Figure 13: **Generated 3D scenes from VIST3A: Wan + AnySplat.** These are 3DGS viewed directly in the interactive viewer. VIST3A preserves high visual quality even under noticeably altered camera trajectories, demonstrating robustness and stability across novel viewpoints.

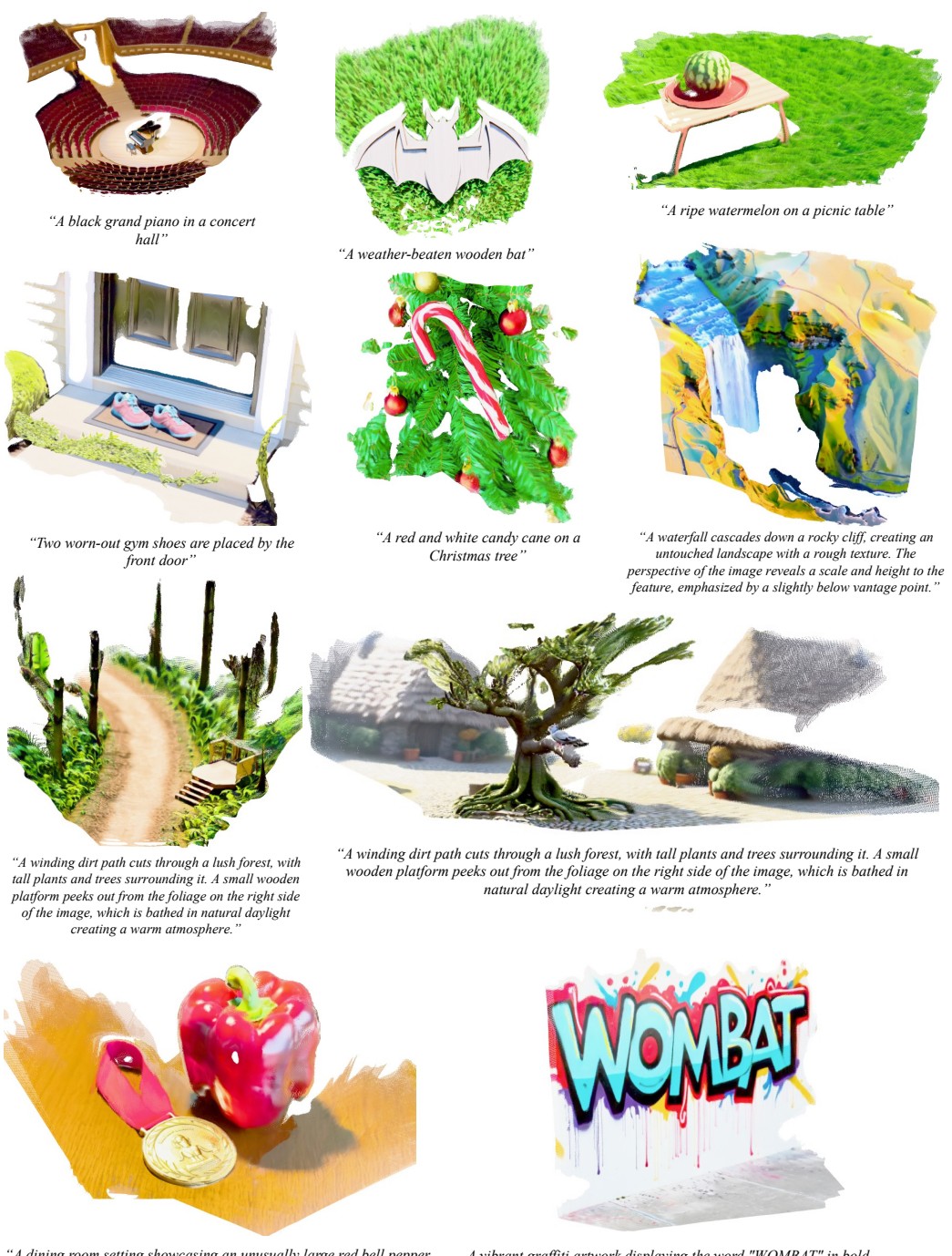

*"A black grand piano in a concert hall"*

*"A weather-beaten wooden bat"*

*"A ripe watermelon on a picnic table"*

*"Two worn-out gym shoes are placed by the front door"*

*"A red and white candy cane on a Christmas tree"*

*"A waterfall cascades down a rocky cliff, creating an untouched landscape with a rough texture. The perspective of the image reveals a scale and height to the feature, emphasized by a slightly below vantage point."*

*"A winding dirt path cuts through a lush forest, with tall plants and trees surrounding it. A small wooden platform peeks out from the foliage on the right side of the image, which is bathed in natural daylight creating a warm atmosphere."*

*"A winding dirt path cuts through a lush forest, with tall plants and trees surrounding it. A small wooden platform peeks out from the foliage on the right side of the image, which is bathed in natural daylight creating a warm atmosphere."*

*"A dining room setting showcasing an unusually large red bell pepper with a shiny, slightly wrinkled texture, prominently placed beside a diminutive golden medal with a red ribbon on a polished wooden dining table. The pepper's vibrant hue contrasts with the medal's gleaming surface. The scene is composed in natural light, highlighting the intricate details of the pepper's surface and the reflective quality of the medal."*

*A vibrant graffiti artwork displaying the word "WOMBAT" in bold, multicolored letters, each character outlined in black to create a striking contrast against the stark white wall. The letters are embellished with various shades of blue, green, red, and yellow, with dramatic splashes of paint scattered around the composition. The texture of the dripping paint adds a dynamic and tactile quality to the mural.*

Figure 14: **Qualitative results on text-to-poinmap generation.** By integrating VGGT, VIST3A generates structurally consistent pointmaps and fine-grained details across diverse prompts.

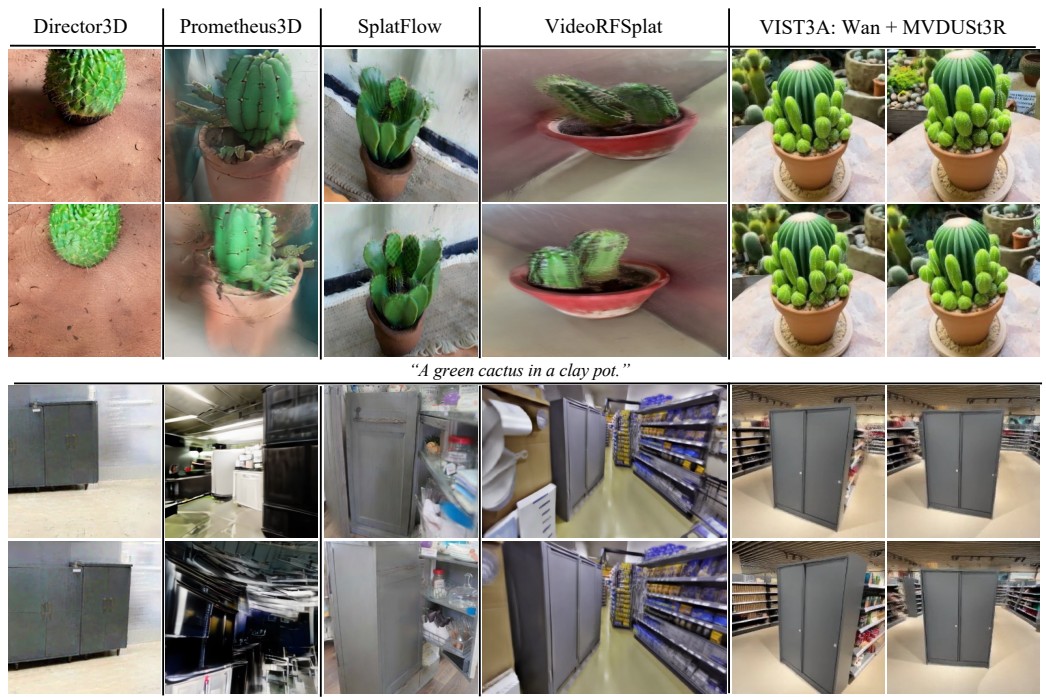

| Director3D | Prometheus3D | SplatFlow | VideoRFSplat | VIST3A: Wan + MVDUSt3R |
|---|---|---|---|---|

*"A green cactus in a clay pot."*

*"The image displays a gray cabinet with closed doors situated next to an open shelf filled with items. The scene appears to be indoors, possibly in a store or office setting."*

Figure 15: **Qualitative comparison of 3DGS generation on SceneBench - VISTA: Wan+MVDUSt3R.**

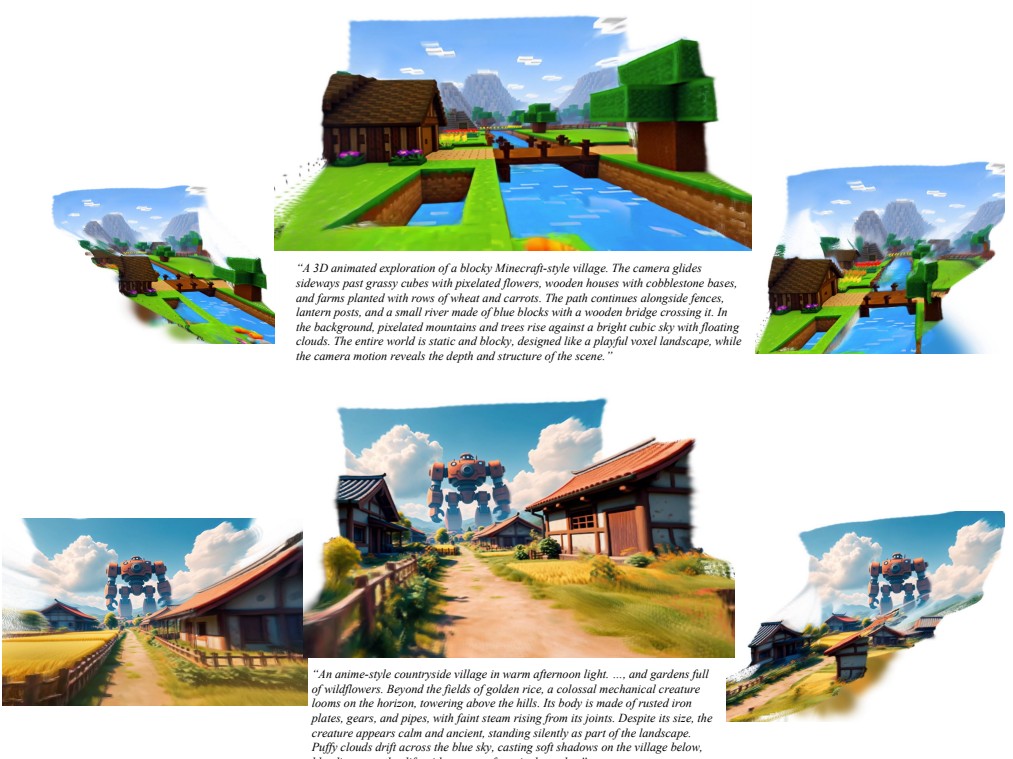

*"A 3D animated exploration of a blocky Minecraft-style village. The camera glides sideways past grassy cubes with pixelated flowers, wooden houses with cobblestone bases, and farms planted with rows of wheat and carrots. The path continues alongside fences, lantern posts, and a small river made of blue blocks with a wooden bridge crossing it. In the background, pixelated mountains and trees rise against a bright cubic sky with floating clouds. The entire world is static and blocky, designed like a playful voxel landscape, while the camera motion reveals the depth and structure of the scene."*

*"An anime-style countryside village in warm afternoon light. ..., and gardens full of wildflowers. Beyond the fields of golden rice, a colossal mechanical creature looms on the horizon, towering above the hills. Its body is made of rusted iron plates, gears, and pipes, with faint steam rising from its joints. Despite its size, the creature appears calm and ancient, standing silently as part of the landscape. Puffy clouds drift across the blue sky, casting soft shadows on the village below, blending everyday life with a sense of magical wonder."*

Figure 16: **Generated 3D scenes from VIST3A: Wan + AnySplat bt extending the number of frames.** These are 3DGS viewed directly in the interactive viewer. VIST3A preserves high visual quality even under noticeably altered camera trajectories, demonstrating robustness and stability across novel viewpoints.

