# OpenReview forum: "Text-to-3D by Stitching a Multi-view Reconstruction Network to a Video Generator"
_ICLR.cc/2026/Conference — ICLR 2026 Oral_

### Official Review · Reviewer_yob1 · 2025-10-21

**Soundness:** 3
**Presentation:** 3
**Contribution:** 3
**Rating:** 8
**Confidence:** 5

**Summary:**

This paper proposes a method for text-guided 3D scene generation. The core idea is to align the intermediate layer features of the video generation model with those of the feedforward 3D reconstruction model, thereby improving the 3D consistency of the video generation model and integrating the two models to complete the generation task. The paper also proposes a post-alignment process to further enhance the overall quality and 3D consistency of the model.

**Strengths:**

1. The motivation of the paper is clear and important. Given that there has been a lot of work on the pre-training phase for 3D scene generation, further optimization and post-training research have become particularly important at this stage.
2. The logic of the paper is relatively sound. The proposed grafting alignment scheme has universality for subsequent research.
3. The paper provides detailed comparison and ablation results. In particular, it has achieved sota performance in text-to-3D scene generation.

**Weaknesses:**

1. Although the grafting scheme proposed in the paper is reasonable, it has some core potential issues:
Because it relies on the alignment of hidden space features, the video model needs to be fine-tuned using multi-view data. This process may impair the generalization ability of the video model.
Since the feedforward reconstruction model itself is also trained on multi-view data, its acceptable distribution is limited, which may also lead to a reduction in the generalization ability of the final model.
Moreover, the capability of the feedforward reconstruction model itself will limit the capability of the entire framework, as they also struggle to successfully model some details (such as grass, branches, etc.). Will this be reflected in the video results of the model?
2. The paper lacks qualitative ablation studies on the alignment step. What puzzles me is that the qualitative results presented in the paper seem to have a tendency of over-saturation. Is this caused by the alignment step? The multi-view data used in the training itself are all from real scenes.

**Questions:**

Please see the questions in the Weaknesses section.

---

> ### Author Response · Authors · 2025-11-21
> **Official Comment by Authors (1)**
>
> We appreciate the insightful feedback. We tried our best to address your concerns and revise the paper accordingly.
>
> ---
>
> ## **Weakness 1**
>
>
> > ### **Weakness 1-1:** *Because it relies on the alignment of hidden space features, the video model needs to be fine-tuned using multi-view data. This process may impair the generalization ability of the video model.*
>
>
> Yes, of course. Video generators are also made to produce dynamic scenes, and are known to have rather weak 3D-consistency. “Finetuning” them for 3D generation means tuning to the narrower domain of static, 3D-consistent scenes. To achieve this, previous works typically finetune only with multi-view datasets, which are by construction 3D-consistent but do not have nearly the diversity of the original video pretraining corpus. Considering that collecting highly diverse multi-view data to prevent degradation during finetuning, we agree with the reviewer that this setting will harm generality - the degraded image quality and weakened prompt adherence are clearly visible for the finetuned model only with the multi-view dataset in Fig. 7 of the (revised) manuscript.
>
>
> Our direct reward finetuning mitigates the problem. It not only aligns the video generator more closely with the stitched decoder, but also improves diversity: Supervision is provided via samples generated by the video model itself, rather than by limited multi-view datasets. Diverse samples from the pretrained video model’s wide domain can therefore be obtained by providing diverse text prompts. Consequently, VIST3A preserves the generality of the video model better, while still enforcing 3D consistency. This contributes to the strong performance on the DPG benchmark, with many long and detailed prompts, giving rise to scenes that one would not find in a normal multi-view dataset.
>
> > ### **Weakness 1-2:** *Since the feedforward reconstruction model itself is also trained on multi-view data, its acceptable distribution is limited, which may also lead to a reduction in the generalization ability of the final model.*
>
> We partially agree with this point. As a framework, VIST3A indeed depends on the acceptable distribution of the feedforward 3D reconstruction model used within it. Our primary target, however, is foundation 3D models that have undergone large-scale, diverse training and thus already cover extremely broad domains. The VGGT and AnySplat models used in our experiments are representative examples of such foundation 3D models. Moreover, progress in 3D reconstruction is extremely rapid: as recent models such as Pi3 [1] and DepthAnythingv3 [2] continue to expand their domain coverage. We emphasize that our framework is generic and designed to work with different 3D backbones, and expect this limitation to be further alleviated over time.
>
> Nevertheless, we agree that it is important to explicitly measure how much the pretrained video generative model’s performance degrades after being adapted through the current VIST3A. To this end, we compare the video generation performance of the original pretrained Wan model and the VIST3A-adapted Wan + AnySplat model using the VBench.
>
> | Model                      | Quality Score | Semantic Score | Total Score |
> |----------------------------|---------------|----------------|------------:|
> | Original video generator   | 0.7827        | **0.7275**         |  0.7716      |
> | VIST3A (Wan + AnySplat)    | **0.8143**        | 0.7143         |  **0.7943**      |
>
>
> As shown above, the VIST3A-adapted model achieves a higher Quality Score and a higher Total Score than the pretrained Wan model, with only a marginal difference in the Semantic Score. This indicates that adapting Wan with VIST3A does not reduce the model’s video generation capability—in fact, its overall VBench performance is slightly improved. We have included this discussion in Appendix E and Table 7. We thank the reviewer for the constructive feedback, which helped us strengthen the clarity of our analysis.

---

> ### Author Response · Authors · 2025-11-21
> **Official Comment by Authors (2)**
>
> > ### **Weakness 1-3:** *Moreover, the capability of the feedforward reconstruction model itself will limit the capability of the entire framework, as they also struggle to successfully model some details (such as grass, branches, etc.). Will this be reflected in the video results of the model?*
>
> We also partially agree with this point. Current feedforward 3D reconstruction models still struggle to capture very fine details, and we acknowledge that this can lead to slightly blurrier renderings when the reconstructed 3D Gaussians are rendered back into images. However, similar to the discussion above, this limitation is continuously being mitigated as 3D reconstruction models rapidly improve.
>
>
> In addition, in our experiments, the limitations of the reconstruction model do not noticeably degrade the final video quality in practice. As shown in the Vbench result, VIST3A built on top of AnySplat achieves a similar semantic score and even improves the quality score compared to the pretrained Wan model. Furthermore, as illustrated in the second (bottom) example of Fig. 7 of the revised manuscript, for challenging structures such as grass and branches, the video outputs of the pretrained Wan model and those of the VIST3A-adapted model remain similar.
>
> ---
>
> ## **Weakness 2: Lack of qualitative ablation on the alignment step & oversaturation**
>
> We thank the reviewer for pointing this out. In the revised manuscript, we have added a qualitative ablation of the alignment step in Fig. 7 in Appendix D.1. Without any finetuning, the pretrained video model often produces temporal motion, which leads to severe ghosting and poor 3D reconstruction quality. When we finetune only on multi-view data, the object motion is suppressed and 3D consistency improves, but the decoded 3D Gaussians become noticeably blurrier and the alignment with the text prompt is weakened. In contrast, our full alignment step produces 3DGS outputs that are both better aligned with the text prompts and sharper in visual quality.
>
>
> Regarding the reviewer’s concern about oversaturation, we observe that the tendency toward slightly oversaturated colors is already present in the pretrained video model itself (see the football example in Fig. 7). Across the variants shown in this figure (no finetuning and our full method), the overall color tone and saturation remain very similar. This suggests that the oversaturation effect originates from the underlying pretrained video generator, rather than being induced by our alignment step. We conjecture that this behavior is related to a broader trend in modern generative video models, which are often trained to favor highly saturated, visually striking outputs.
>
> ---
>
> **References**
>
> [1] Wang et al. π3: Permutation-Equivariant Visual Geometry Learning, arXiv 2025.
>
>
> [2] Lin et al. Depth Anything 3: Recovering the Visual Space from Any Views. arXiv 2025.

---

> > ### Comment · Reviewer_yob1 · 2025-11-26
> >
> > I appreciate the authors' detailed reply. Most of my concerns have been clarified. Regarding the issue of over-saturation, may I ask what resolution you used for video generation, and is it consistent with the generation resolution of the original video model?

---

> > > ### Author Response · Authors · 2025-11-26
> > >
> > > We sincerely appreciate your effort in carefully reading our response and the revised manuscript. We are very happy to hear that most of your concerns have been clarified.
> > >
> > >  Regarding your question about the resolution: in our experiments, we did not use the native resolutions (832×480 and 1280×720) supported by Wan 2.1. Instead, we generated videos at a lower resolution because the AnySplat backbone operates on lower-resolution inputs, and using a lower resolution makes the overall system more efficient. In principle, one could generate videos at the native resolution of Wan 2.1 and then apply appropriate interpolation when feeding the stitched 3D model, but this would introduce significant computational overhead into the pipeline.
> > >
> > > We hope this addresses your remaining question and leaves you with an even more positive view of our work.

---

> ### Comment · Reviewer_yob1 · 2025-11-26
>
> Based on my own experience, I suspect this might be the reason making the visual effects overly saturated. I encourage the authors to try aligning the resolution and frame number in future attempts, or to apply the method proposed in the paper after fine-tuning the video model under specific resolutions and frame numbers.
>
> Because the pipeline proposed in this paper is very important and provides insights into the integration of subsequent video foundation models and 3D foundation models, I decide to keep my score for acceptance.
>
> At the same time, I also encourage the authors to continue exploring the extension of the framework in dynamic scene generation.

---

> > ### Author Response · Authors · 2025-11-27
> >
> > We sincerely thank you for your insightful suggestions and for proposing multiple promising future research directions. In particular, we fully agree that extending the framework toward dynamic 4D scene generation is a highly valuable and exciting direction for future work.
> >
> > Regarding your comment on the potential cause of the overly saturated visual effects, we would like to note that, as shown in Figure 15 of Wan 2.1 [1], a similar level of saturation is also observed. We believe that additional factors may contribute to this phenomenon, such as the fact that pre-trained video models are often trained primarily on visually striking video content, as well as the model’s strong dependence on the CFG.
> >
> > Finally, we sincerely appreciate your recognition of the importance of our proposed pipeline and your positive assessment of its potential impact on the integration of video and 3D foundation models. While we fully respect your judgment, if you feel that our rebuttal has sufficiently addressed your concerns, we would be grateful if you might consider a stronger recommendation.
> >
> >
> > ### References
> >
> > [1] Team Wan. WAN : OPEN AND ADVANCED LARGE-SCALE VIDEOGENERATIVE MODELS. arXiv 2025.

---

### Official Review · Reviewer_3u1z · 2025-10-24

**Soundness:** 3
**Presentation:** 4
**Contribution:** 3
**Rating:** 8
**Confidence:** 4

**Summary:**

The paper introduces an approach to replace the decoder in a video diffusion model with a pre-trained feed-forward 3D regression network, such as one that can perform pixel-oriented Gaussian splatting or 3D point map prediction. The resulting model can then generate 3D outputs directly, for instance from textual prompts.

The method repurposes earlier techniques for networks stitching. The starting point is a video generator based on latent diffusion. This uses a variational autoencoder to encode and decode RGB frames into a latent space. The idea is to "stitch" (by means of a linear adapter) the given pre-trained 3D regressor as a decoder replacement, thus obtaining a new decoder that outputs a 3D representation of the generated scene instead of raw pixels. Stitching identifies the subset of layers in the regressor that are best replaced by the encoder, selecting this as a stitching point. Notably, the resulting encoded+stitched decoder can also function as a replacement for the original regressor, and in some cases outperforms it.

Given the new encoder-decoder, the diffusion model is then jointly fine-tuned with it. This step is necessary to ensure the old diffusion model and new encoder-decoder work well together. This step uses direct reward fine-tuning, where the diffusion process is unrolled and the output is assessed using image-based metrics, which are then optimised.

**Strengths:**

* The idea of using network stitching to add a deterministic 3D regression head to a multi-view image generator is creative, practical, and, empirically, quite effective. I think it is definitely worth sharing it with the community.

* The paper also proposes a practical manner to re-adjust the diffusion process in the generator to adapt it to the new encoder-3D-decoder. The latter uses reward fine-tuning.

**Weaknesses:**

* Perhaps with the exception of direct reward fine-tuning, the model is not really learning a 3D latent space, but rather to decode a multi-view latent space to 3D directly. While multi-view is close to 3D, it may miss some important properties that a native 3D latent space may capture better, the most obvious one being extent. By this, I mean that the part of the 3D world which is reconstructed is commensurate to the part of the 3D world captured in the multiple views, which in general do not provide full coverage and may leave the final 3D reconstruction incomplete. This is pretty obvious from the “holes” in figure 1.

* It is not clear to me how the underlying video generator decides to select the viewpoints that are (implicitly) generated. Likewise, it is not clear what happens with dynamic contents.

* Eq. (2) measures the similarity of feature alignment to decide where to cut. However, the experiments in Section 4.3 suggest the obvious alternative to measure the quality of the new VAE in terms of 3D reconstruction in order to perform this selection. Why is the first option preferred?

* I was a little surprised to notice that there is no supplementary material. Videos of the results would have gone a long way in illustrating their quality.

**Questions:**

* See above, what are the limitations of using a multi-view latent space instead of learning a “proper” 3D one?

* How do the various image generators choose which viewpoints to (implicitly) consider for generation?

* How do you handle potential dynamic objects in the generated scene?

* Will code be released?

---

> ### Author Response · Authors · 2025-11-21
> **Official Comment by Authors (1)**
>
> We are grateful for the reviewer’s thoughtful questions and comments, which are valuable not only for the present paper but also for our ongoing/future work.
>
> ---
>
> ### **Weakness 1 & Question 1: Limitations of multi-view latent space**
>
> Thank you for bringing this up. We agree that learning a native 3D latent space is an interesting and promising future direction. In principle, it could more comprehensively capture the full extent of the scene and reduce occlusion-related gaps.
> In the current visual AI landscape, several bottlenecks impede native 3D generative modelling compared to our multi-view latent scheme. So far, native 3D models have largely been restricted to object-centric generation [1, 2]. When extended to scenes, they struggle with diversity and robustness [3]. The limiting factors appear to be
> 1. Data: Constructing large datasets of complete 3D scenes without “holes” is challenging. At the same time, the generality of modern generative models largely comes from access to huge, diverse training sets. The scarcity of high-quality 3D scans is a critical limitation for learning native 3D representations. Even in very recent work [3], the training data is mostly restricted to indoor scenes. In contrast, the data that underpins generative video models is extremely large and diverse. This sheer data volume is, in our view, a key reason for the high-quality results of  VIST3A and its ability to nevertheless produce convincing 3D outputs for a wide variety of prompts.
>
> 2. Availability of off-the-shelf models. Another advantage of our design is that it leverages some of the most impressive successes of recent visual AI. For both video generation and 3D reconstruction, powerful models have become a commodity. In contrast, native 3D generators are an exciting research field, but off-the-shelf tools with the maturity of WAN or VGGT are not yet available, and one would have to build the entire stack from scratch, which is precisely what we try to avoid.
>
> A native 3D generative model would, in an ideal world, be the most principled and elegant solution - but we don’t see a way to practically implement it with reasonable effort. We see our multi-view latent scheme as a “next-best solution” given the current tools. It might one day be superseded by a native 3D model - but it is at this point unclear how long that might still take.
>
> More pragmatically, the “holes” in our current reconstructions could be mitigated with available technology, i.e., inpainting (respectively, scene completion) - an interesting direction for future work.
>
>
> ---
>
> ### **Weakness 2-1 & Question 2: How viewpoints are determined**
>
> You have raised an important point. We will make sure to discuss it more carefully.
>
> In our current setup, viewpoints are controlled through the text prompt. State-of-the-art video models such as Wan 2.x are quite good at interpreting and following viewpoint instructions (like *“aerial view,” “pan left to right”*). Since we built on Wan 2.1, VIST3A inherits this property.  In the revised paper, we visualize this behavior in Fig. 10 in Appendix E.
>
>
> We emphasize that our framework is generic and designed to work with different video diffusion backbones; including models that support viewpoint/camera control. In particular, one could equally stitch the 3D decoder to a camera-conditioned video model like [4] to control the viewpoints explicitly via camera matrices - an avenue we are currently exploring.
>
> ---
>
> ### **Weakness 2-2 & Question 3: Dynamic scene**
>
> 3D reconstruction models used in VIST3A are designed for static scenes; dynamic objects violate their underlying assumptions. In our case, the stitched decoder reconstructs the moving object at every location it appears in the video, so the 3D representation becomes cluttered with “stop-motion copies” along the trajectory of a moving object, see Fig. 7 in our revised paper. As the proposed reward finetuning explicitly penalizes outputs that are implausible and not 3D-consistent, it eliminates these artifacts (but of course does not reconstruct dynamic scenes - that would need a suitable 4D reconstruction model).
>
> ---
>
> ### **Weakness 3: Stitching layer selection with Eq. 2**
>
>
> You are absolutely right. Exhaustively searching for the stitching layer that maximises overall performance would be ideal. Unfortunately, this would be extremely expensive, as one would have to run finetuning for a substantial number of candidate layers (e.g., 24 encoder layers of VGGT), sacrificing an important advantage of our stitching method.
>
>
> Fortunately, we find that a low linear stitching error (according to our MSE metric) turns out to correlate well with the final performance of the stitched model. We see an efficient proxy for finding the stitching layer as a key component to make our model practically useful.

---

> ### Author Response · Authors · 2025-11-21
> **Official Comment by Authors (2)**
>
> ---
>
> ### **Weakness 4: Supplementary material**
>
> We apologize for the confusion regarding the supplementary material. An interactive 3D viewer offers a more tangible way to appreciate and assess the 3D scenes produced by VIST3A. Therefore, we opted for a link to an anonymized GitHub repository (in the abstract), from which the anonymized project page can be accessed, rather than a file upload. Besides the 3D viewer, the webpage also contains videos of reconstructed scenes. For easier access, we have also updated the revised paper so that the link now directly points to the anonymized project page instead of the GitHub repository.
>
>
> ---
>
>
> ### **Question 4: Code release**
>
> Yes, we will release the code upon acceptance - we are eager to share our codebase and promote model stitching as a general strategy for combining foundational vision models.
>
>
> ---
>
> **References**
>
> [1] Xiang et al. Structured 3d latents for scalable and versatile 3d generation. CVPR 2025
>
> [2] Lan et al. Gaussiananything: Interactive point cloud latent diffusion for 3d generation. ICLR 2025.
>
> [3] Huang et al. Terra: Explorable Native 3D World Model with Point Latents. arXiv 2025.
>
> [4] Bahmani et al. AC3D: Analyzing and Improving 3D Camera Control in Video Diffusion Transformer. CVPR 2025.

---

### Official Review · Reviewer_HSYK · 2025-10-27

**Soundness:** 4
**Presentation:** 3
**Contribution:** 3
**Rating:** 8
**Confidence:** 4

**Summary:**

This paper presents VIST3A, a framework for text-to-3D scene generation that combines the power of modern latent text-to-video models with geometric abilities of recent feedforward 3D reconstruction models through a model stitching mechanism. The authors first identify the layer in the 3D reconstruction model that best matches the latent representation produced by the encoder of video VAE, and stitch the two parts via a lightweight stitching layer. Then, they apply direct reward finetuning to align the generative model's latent distribution with the stitched 3D decoder. VIST3A achieves strong quantitative and qualitative results on multiple benchmarks, outperforming recent text-to-3D Gaussian splatting models. VIST3A also exhibits promising qualitative performance on text-to-pointmap generation.

**Strengths:**

* Conceptual originality: first to demonstrate high-fidelity text-to-3D generation by connecting video generator and feedforward 3D reconstructor through model stitching.
* Efficient training: the framework uses self-supervised alignment and reward tuning, reducing data requirements.
* Modular and generalizable: the authors demonstrate the framework's feasibility across multiple video generators and 3D reconstructors, showing the generalizability of proposed method.
* Superior experimental performance: VIST3A achieves superior scores on diverse benchmarks along with fancy qualitative results, exhibiting stable outperforming compared to baseline methods.

**Weaknesses:**

* Limited theoretical justification: while empirically compelling, the choice of layer for stitching is primarily heuristic (minimizing MSE) and lacks deeper analysis of representation compatibility.
* Concerns about reward finetuning complexity: DRT requires full denoising loop and multi-view rendering for each step, introducing heavy computational overhead. The scalability to larger video models or higher resolutions may be challenging.
* No human evaluation: subjective realism and prompt alignment are only indirectly assessed via CLIP/HPSv2 metrics.

**Questions:**

* It is mentioned in line 261 that "As we keep the encoder frozen, the generated latents can be decoded by the original video decoder to obtain multi-view images", however, the latents produced by the diffusion model would have changed along the reward tuning, how could the latents be constantly decoded to multi-view images using the original decoder?
* There seems lack descriptions for several notations in their first appearance. For example, $D_E$ and $D_F$, which are the dimensions of latents $B$ and activations $A$ respectively from line 223.

---

> ### Author Response · Authors · 2025-11-21
> **Official Comment by Authors (1)**
>
> We appreciate the insightful comments.  We have incorporated the suggested changes; your feedback was very helpful to refine our paper.
>
> ---
>
> ### **Weakness 1: Limited theory and analysis of representation compatibility.**
>
>
> We agree: What works in practice in visual AI is a largely empirical question, and our primary focus was to experimentally demonstrate that a range of video VAEs and feedforward 3D models are stitchable (even with a simple MSE-based criterion). Still, it is true that complementary theoretical insights are desirable where possible. For stitching, there is some hope:
>
> - **1. Theory:** Theoretical aspects of the stitching approach have been rigorously studied in the literature. According to Theorem 1 in [1], the stitching error—defined as the error of the hybrid network formed by connecting the source model’s early layers ($f_1$) to the target model’s latter layers ($g_2$)—is upper-bounded by the MSE of the stitching layer. Specifically, the proof shows:
>
>     $$\\mathbb{E} [ \\| g_2(S_{1,2}f_1)(x) - g_2(f_2)(x) \\|^2 ] \\leq \\kappa_2^2 \\underbrace{\\mathbb{E} [ \\| S_{1,2}f_1(x) - f_2(x) \\|^2 ]}_{\\text{MSE}}$$
>
>     where $\kappa_2$ is the Lipschitz constant of $g_2$ (representing the subsequent layers of the target model), $S_{1,2}$ is the linear stitching transformation, and $f_1$ and $f_2$ denote early layers of the source and target models, respectively. Thus, an MSE is related to the stitching error, supporting our use of MSE. We included this discussion in Section 4.4 of our revised manuscript. Thank you for your constructive feedback.
>
> - **2. Deep analysis of representation compatibility:** Moreover, [1] shows that the term on the right-hand side of the inequality shares a similar form with kernel alignment. To explore this, we compared CKA [3] with our MSE metric and illustrated the results in Fig. 6 in the revised manuscript. While CKA captures the general performance trend, it lacks the precision to identify the optimal layer (in the example, layer 2). In contrast, our MSE metric accurately pinpoints the best layer. We conclude that directly utilizing the MSE is more reliable than generic similarity metrics. We also included this discussion in Section 4.4 of our revised manuscript. We appreciate your helpful feedback.
>
> ---
>
> ### **Weakness 2: Computational overhead of direct reward finetuning**
>
>
> We appreciate the reviewer’s interest regarding the computational aspects of our method. To clarify the impact, we decompose the computational overhead into memory usage and wall-clock time:
>    - **1. Memory Usage**: While, naturally, our reward tuning requires more GPU memory than multi-view finetuning, it is quite manageable and not a fundamental limitation. In our implementation, the overhead from long gradient paths is controlled via activation/gradient checkpointing, which proved sufficient for training at the 14B-parameter scale (so, we already succeeded in scale-up). Furthermore, standard distributed strategies like ZeRO or FSDP can be readily incorporated to further reduce the per-GPU footprint if needed.
>    - **2. Wall-Clock Time**: In fact, direct reward tuning is not the bottleneck when it comes to overall training efficiency. See the ablation study in the Appendix, where we compare multi-view finetuning and our reward tuning with the same budget of GPU-hours. With equal training time, reward finetuning outperforms multi-view finetuning, i.e., reward finetuning can reach the same performance faster in terms of wall-clock time. We attribute this to the very nature of direct reward tuning: by utilizing gradients to directly optimize the model, they achieve rapid alignment, as also observed in [1].
>
>
> Overall, we find that the computational overhead is not problematic: memory usage is manageable, and training time is rather modest by the standards of vision foundation models. On the contrary, direct reward finetuning offers fundamental advantages compared to multi-view finetuning:
> - **1. Strong Supervision Signal**:  Unlike generative losses used with multi-view datasets, our method directly leverages supervision from the 3D representation space, which yields improved 3D consistency and visual quality.
> - **2. Supervision with Diverse Domain:** Finetuning with multi-view data is limited by data availability; obtaining a truly diverse multi-view dataset is difficult, and using datasets with limited diversity will overtune the video generator to a narrow domain. In contrast, direct reward finetuning utilizes the pretrained video model itself as a data generator. Simply feeding diverse text prompts ensures that the training samples are also diverse. As the model is optimized via rewards on these samples, it maintains its generality.

---

> ### Author Response · Authors · 2025-11-21
> **Official Comment by Authors (2)**
>
> ---
>
> ### **Weakness 3: Lack of human evaluation**
>
> Indeed, it makes sense to complement the quantitative evaluation with human preference scores. Accordingly, we have conducted a user study, where we rendered videos from generated 3DGS models and asked participants to rank five methods (VIST3A, Director3D, SplatFlow, Prometheus3D, VideoRFSplat) based on two criteria: Prompt Alignment and Visual Quality. A total of 28 participants evaluated 14 randomly selected samples.
>
> As shown in the table below, VIST3A achieved the lowest average rank on both criteria by some margin. Notably, participants rank VIST3A as the 1st choice in 68.68% of cases for text alignment and 87.91% for visual quality. We acknowledge the small scale of the study due to limited time. We will be happy to expand it for our revision and have included them in Appendix D.3.
>
> | Method | Text Alignment (Average Rank $\downarrow$) | Visual Quality (Average Rank $\downarrow$) |
>  | :--- | :---: | :---: |
> | Director3D | 3.03 | 2.99 |
>  | SplatFlow | 3.38 | 3.88 |
> | Prometheus3D | 3.25 | 3.71 |
> | VideoRFSplat | 2.74 | 2.92 |
>  | **VIST3A (Wan + AnySplat) (Ours)** | **1.54** | **1.45** |
>
> ---
>
> ### **Question 1: Decoding via original decoder**
>
> We may not have explained this in enough detail. One of our rewards - the multi-view image quality reward -  is applied after decoding the latents into image frames with the original video decoder. Consequently, reward tuning explicitly encourages the model to stay compatible with the original video decoder: latents that are mapped into higher-quality images by the original decoder are rewarded, whereas latents that decode poorly are discouraged.
>
> ---
>
> ### **Question 2: Missing notations**
>
> Thank you for reading so carefully. We have revised Step 1 in Section 3.1 to introduce all relevant symbols upon their first appearance.
>
> ---
>
> **References**
>
> [1] Insulla et al. Towards a Learning Theory of Representation Alignment. ICLR 2025.
>
> [2] Prabhudesai et al. Video Diffusion Alignment via Reward Gradients. arXiv 2024.
>
> [3] Kornblith et al. Similarity of neural network representations revisited. ICML 2019

---

> > ### Comment · Reviewer_HSYK · 2025-11-26
> >
> > Thanks for the authors' detailed rebuttal. Most of my concerns have been addressed. I decide to maintain my score.

---

> > > ### Author Response · Authors · 2025-11-27
> > >
> > > We are pleased to hear that our rebuttal has addressed most of your concerns, and we sincerely appreciate your decision to maintain your score. We are also very grateful for your constructive comments and questions, which have played an important role in improving our manuscript.
> > >
> > > Once you have completed the evaluation of your full review batch, if you feel it is appropriate, we would be thankful if you could consider a stronger recommendation. In any case, we greatly appreciate your time and effort and fully respect your final decision.

---

### Official Review · Reviewer_BZA7 · 2025-10-28

**Soundness:** 3
**Presentation:** 3
**Contribution:** 3
**Rating:** 8
**Confidence:** 4

**Summary:**

This paper presents VIST3A, a novel and effective framework for text-to-3D generation. Its core contribution is a model stitching approach that seamlessly connects a pre-trained text-to-video generator to a powerful, pre-trained feedforward 3D reconstruction model, repurposing the latter as a strong 3D VAE decoder. This design elegantly leverages the geometric prior of state-of-the-art 3D vision models without requiring their expensive replication through large-scale training. To ensure alignment between the generative model and the stitched decoder, the authors further introduce a direct reward fine-tuning strategy. This technique aligns the denoising process to produce latents that are both 3D-consistent and within the input domain of the decoder. The experiments demonstrate superior performance compared with baseline methods.

**Strengths:**

1. Novel and generalizable framework. The proposed model stitching method is a key innovation. It provides a cost-effective way to harness the power of pre-trained 3D vision models as strong decoders, eliminating the need for expensive training from scratch. Its generality is demonstrated by successfully combining different video encoders with different 3D models, enabling the generation of both 3D Gaussian splats and pointmaps.
2. Thorough experiments. The paper provides compelling evidence through extensive experiments. It robustly benchmarks multiple model combinations against strong baselines. It also includes well-designed ablations that critically validate core design choices, specifically the stitching layer selection and the contribution of direct reward fine-tuning.

**Weaknesses:**

While the stitching itself is data-efficient, the subsequent direct reward fine-tuning is computationally intensive. It requires unfolding the entire denoising trajectory and backpropagating through multiple decoders and reward models, which is costly in both time and GPU memory.

**Questions:**

1. The paper demonstrates successful stitching across various model pairs. Could you elaborate on the scenarios or conditions where your stitching approach might fail? For instance, have you encountered pairs of video VAEs and 3D models where no suitable stitching layer could be found, and what might be the underlying reasons (e.g., architectural mismatch, vastly different representation spaces)?
2. The performance of VIST3A is inherently tied to the capabilities of the chosen 3D foundation model. What are the key properties or criteria you considered when selecting a model like AnySplat or VGGT for stitching? For instance, how do the specific 3D representations (Gaussians vs. pointmaps), architectural design, or training data of the foundation model influence the final text-to-3D capabilities of VIST3A, and are these selection criteria generalizable to future models?(Note: This question seeks your insight and perspective, not additional experiments.)

---

> ### Author Response · Authors · 2025-11-21
> **Official Comment by Authors (1)**
>
> Thank you for your valuable feedback on our paper. We have made every effort to address your comments and improve the manuscript accordingly.
>
>
> ---
>
>
> ### **Weakness 1: Computational overhead of direct reward finetuning.**
>
> We appreciate the reviewer’s interest regarding the computational aspects of our method. To clarify the impact, we decompose the computational overhead into memory usage and wall-clock time:
> - **1. Memory Usage**: While, naturally, our reward tuning requires more GPU memory than multi-view finetuning, it is quite manageable and not a fundamental limitation. In our implementation, the overhead from long gradient paths is controlled via activation/gradient checkpointing, which proved sufficient for training at the 14B-parameter scale. Furthermore, standard distributed strategies like ZeRO or FSDP can be readily incorporated to further reduce the per-GPU footprint if needed.
> - **2. Wall-Clock Time**: In fact, direct reward tuning is not the bottleneck when it comes to overall training efficiency. See the ablation study in Table 5 in the Appendix, where we compare multi-view finetuning and our reward tuning with the same budget of GPU-hours. With equal training time, reward finetuning outperforms multi-view finetuning, i.e., reward finetuning can reach the same performance faster in terms of wall-clock time. We attribute this to the very nature of direct reward tuning: by utilizing gradients to directly optimize the model, they achieve rapid alignment, as also observed in [1].
>
>
> Overall, we find that the computational overhead is not problematic: memory usage is manageable, and training time is rather modest by the standards of vision foundation models. On the contrary, direct reward finetuning offers fundamental advantages compared to multi-view finetuning:
> - **1. Strong Supervision Signal**:  Unlike generative losses used with multi-view datasets, our method directly leverages supervision from the 3D representation space, which yields improved 3D consistency and visual quality.
> - **2. Supervision with Diverse Domain:** Finetuning with multi-view data is limited by data availability; obtaining a truly diverse multi-view dataset is difficult, and using datasets with limited diversity will overtune the video generator to a narrow domain. In contrast, direct reward finetuning utilizes the pretrained video model itself as a data generator. Simply feeding diverse text prompts ensures that the training samples are also diverse. As the model is optimized via rewards on these samples, it maintains its generality.
>
>
>
> ---
>
> ###  **Question 1: Elaborate on scenarios where stitching might fail.**
>
>
> We appreciate this insightful question. Potential failure modes are indeed important.
>
>
> Our method relies on the assumption that the VAE latent and some intermediate representation of the 3D model are sufficiently compatible under a ***linear*** transformation. For video VAEs and 3D models trained on large-scale data, we find that this assumption is tenable, and a suitable stitching layer can be identified. We attribute this to (1) foundation-scale training with the large-scale data, leading to overlapping domains where both models operate well, and (2) the nature of VAE latents, which are rather low-level and thus expected to be similar across vision models.
>
> From these considerations, we can hypothesize scenarios where stitching might fail. (1) if the VAE and the 3D model are trained on small, disjoint datasets or on different, restricted domains, then they may no longer share a feature space and may be unable to communicate via a linear layer (as extreme examples, imagine exotic image modalities like medical scans or microscopy). Condition (2) could be violated if the architecture of the 3D model somehow skips low-level vision, i.e., its early layers do not first build up a 2D representation before gradually transitioning to 3D reasoning. We are not aware of any such model, but we anticipate that, if it exists, it may not be suitable for stitching to a video model whose output frames reside in 2D image space.

---

> ### Author Response · Authors · 2025-11-21
> **Official Comment by Authors (2)**
>
> ---
>
> ###  **Question 2: Key properties or criteria for selecting models.**
>
> True, we may not have explained well enough what motivated our choices.
>
> For 3D models, a main criterion was the scale of the pretraining dataset, which determines the generality of a foundation model. The assumption was that the generative video model covers a larger and more diverse set of scenes, so the 3D model would be the limiting factor. Therefore, we prioritized models trained on very large datasets. At the time of submission, AnySplat was the most generalizable public model for Gaussian Splats that we were aware of. Similarly, VGGT was the pointmap model that could handle the widest variety of scenes. MV-DUSt3R was chosen as a lightweight alternative, not least to facilitate quick development.
>
> Regarding the type of representation, 3D Gaussian Splatting (3DGS) is more suitable for generating visual assets, due to its superior novel-view rendering. However, we intentionally included pointmap models (MV-DUSt3R, VGGT) to test and demonstrate that the stitching approach is universal and not limited to a specific representation.
>
> We believe these considerations remain valid for the future. We anticipate that stitching will naturally become more reliable as video models continue to ingest ever more data, all but eliminating the risk of disjoint domains (c.f. Q1). At least in the short run, the bottleneck for the final performance is more likely the 3D reconstruction model, for which it is more difficult to assemble a truly foundation-scale dataset.
>
> We have included the discussion in Appendix C.3. Thank you for the constructive feedback, we feel it has strengthened our paper.
>
> ---
>
> **References**
>
> [1] Prabhudesai et al. Video Diffusion Alignment via Reward Gradients. arXiv 2024.

---

> > ### Comment · Reviewer_BZA7 · 2025-11-27
> > **Thanks for rebuttal**
> >
> > I'd like to thank the authors for their rebuttal which addresses most of my concerns. I decide to maintain my positive score.

---

> > > ### Author Response · Authors · 2025-11-27
> > >
> > > We are very glad to hear that our rebuttal has addressed most of your concerns, and we sincerely thank you for maintaining your positive score. We also truly appreciate your constructive comments and questions, which have been extremely helpful in improving our manuscript.
> > >
> > > After you have finished evaluating your full review batch, if you happen to feel it is appropriate, we would be grateful if you could consider a stronger recommendation. In any case, we deeply appreciate your time and effort and fully respect your final decision.

---

### Public Comment · ~Dany_Li1 · 2025-11-20
**Inquiry: video decoder during training pipeline**

I appreciate the reasonable motivation, novel technique and insightful designs in this paper. I raise this comment as I am confused by how the video decoder is used during the training pipeline. As seen in Figure 3, the video DiT provides a latent. This latent is further fed into the stitched reconstruction model and the video decoder, which are used to calculate the 3D consistency reward. This thinking behind is novel and solid, but I am confused by how training pipeline proceeds:

1. Do you feed a noisy video latent into the video decoder? Commonly, multiple denoising steps are proceeded during inferencing a video diffusion. In the proposed training pipeline, it seems that latent is processed by the DiT once, yielding a noisy latent with only one step denoise process. How does the video decoder decode a valid video from this noisy latent?

2. If not, do you denoise multiple steps to get the clean video latent during training? This raises a concern on the training efficiency. What is the time cost for a full denoising during training?

Thanks so much for your time and patience! I have learned a lot from your excellent work and this question really confuses me. I believe your insightful reply helps with our better understanding.

---

> ### Author Response · Authors · 2025-11-23
>
> Dear Dany Li,
>
> We appreciate your interest in our work. Our alignment training algorithm is presented in Algorithm 1 in Appendix B.2, and we hope it helps clarify the full training pipeline. Below are detailed answers to your questions:
>
> - One step: In our alignment-training sampling process, we do not use 1-step sampling, since decoding from a single denoising step can yield overly noisy latents and lead to incorrect generation outcomes. Instead, to reduce training cost, we use fewer denoising steps than inference, but still within a range where the generated results remain feasible for computing the reward.
> - Training time: For a fair comparison in terms of training time, we match the GPU-hours between our method and the multi-view finetuning baseline. Under this equal-compute setup, our alignment training outperforms multi-view finetuning, showing our training-time efficiency. For more details, please refer to our responses to Reviewer BZA7 (Weakness 1) and Reviewer HSYK (Weakness 2), where we provide additional clarification.
>
> We hope that this answer resolves your questions.
>
> Sincerely,
>
> Authors

---

> > ### Public Comment · ~Dany_Li1 · 2025-11-24
> >
> > Thanks for your detailed explanation and I have learned a lot from it. I believe this training paradigm is reasonable and effective. Thanks again for your excellent work.

---

### Meta-Review · Area_Chair_iDrK · 2026-01-08

**Summary:**

This paper received unanimously positive reviews. The main concerns raised in the reviews are:
1. computational cost in the direct reward fine-tuning step (`BZA7`, `HSYK`).
2. inadequate discussion on failure cases (`BZA7`).
3. limited theoretical justification (`HSYK`).
4. missing human evaluation (`HSYK`).
5. potential gaps from multi-view to full 3D reconstruction (`3u1z`).
6. missing qualitative ablations (`yob1`).

Overall, this is a very solid paper proposing a simple and effective technique using _model stitching_ to repurpose video models and 3D reconstruction models for text-to-3D generation. All reviewers agree that this is an innovative and effective approach and provides interesting insights.

**Reviewer Concerns:**

Three reviewers (`BZA7`, `HSYK`, `yob1`) have indicated that their concerns have been addressed, and I believe the detailed responses have also addressed most of the concerns raised by the remaining reviewer (`3u1z`).

**Reviewer Scores:**

1. Reviewer `BZA7` (8->8): the reviewer indicated that they would maintain the positive rating.
2. Reviewer `HSYK` (8->8): the reviewer indicated that they would maintain the positive rating.
3. Reviewer `3u1z` (8->8+): it is likely that the reviewer would maintain or increase the rating.
4. Reviewer `yob1` (8->8): the reviewer indicated that they would maintain the positive rating.

---

### Decision · Program_Chairs · 2026-01-26

Accept (Oral)